

# A global dataset of standardized moisture anomaly index incorporating snow dynamics from 1948 to 2010

Lei Tian[1,2], Baoqing Zhang[2], Pute Wu[3]

[1]Institute of Green Development for the Yellow River Drainage Basin, Lanzhou University, Lanzhou, 730000, China
[2]Key Laboratory of Western China's Environmental Systems (Ministry of Education), College of Earth and Environmental Sciences, Lanzhou University, Lanzhou, 730000, China
[3]Institute of Soil and Water Conservation, Northwest A&F University, Yangling, 712100, China

*Correspondence to*: Baoqing Zhang (baoqzhang@lzu.edu.cn)

**Abstract.** Drought indices are hard to balance in terms of versatility (effectiveness for multiple types of drought), flexibility of timescales, and inclusivity (to what extent they include all physical processes). A lack of consistent source data increases the difficulty of quantifying drought. Here, we present a global monthly drought dataset from 1948 to 2010 based on a multitype and multiscalar drought index, the standardized moisture anomaly index incorporating snow dynamics ($SZI_{snow}$), driven by systematic fields from an advanced data assimilation system. The proposed $SZI_{snow}$ dataset includes different

physical water–energy processes, especially snow processes. Our evaluation of the dataset demonstrates its ability to distinguish different types of drought across different timescales. Our assessment also indicates that the dataset adequately captures droughts across different spatial scales. The consideration of snow processes improved the capability of $SZI_{snow}$, and the improvement is particularly evident over snow-covered high-latitude (e.g., Arctic region) and high-altitude areas (e.g., Tibetan Plateau). We found that 59.66% of Earth's land area exhibited a drying trend between 1948 and 2010, and the

remaining 40.34% exhibited a wetting trend. Our results also show that the $SZI_{snow}$ dataset successfully captured the large-scale drought events that occurred across the world; there were 525 drought events with an area larger than 500,000 km² globally during the study period, of which nearly 70% had a duration longer than 6 months. Therefore, this new drought dataset is well suited to monitoring, assessing, and characterizing drought, and can serve as a valuable resource for future drought studies.

# 1 Introduction

Drought is one of the most costly and complex natural hazards, commonly causing significant and widespread adverse impacts on many sectors of society (Aghakouchak et al., 2015; He et al., 2020). The severity, extent, and duration of drought are likely to intensify across the world under the effects of climate change (Ault, 2020; Mann and Gleick, 2015). There has been increasing global interest in measures to improve the capability of drought quantification and various drought indices

have been proposed over the past several decades (Liu et al., 2018; Esfahanian et al., 2017). However, current drought





indices struggle to reconcile versatility (ability to quantify multiple types of drought), flexibility of temporal scale (effective across different timescales), and comprehensiveness (to what extent they include all hydrological processes). Additionally, these drought indices are derived from multifarious data sources, rather than systematic and consistent physical data from the same source (Ahmadalipour and Moradkhani, 2017; Hoffmann et al., 2020). As a result, different sectors of society have

rarely collaborated to synergistically fight against drought using a comprehensive drought index.

The propagation of drought is related to changes in numerous interconnected variables of hydrometeorological processes (e.g., precipitation, evapotranspiration, streamflow, and soil moisture). Yet a major portion of currently available drought indices focus on only one aspect of drought evolution. For example, the Rainfall Anomaly Index (RAI; Zhu et al., 2021), Streamflow Drought Index (SDI; Nalbantis and Tsakiris, 2009), and Soil Moisture Deficit Index (SMDI; Narasimhan and

Srinivasan, 2005) focus only on precipitation (meteorological drought), streamflow (hydrological drought), and soil moisture (agricultural drought), respectively (Fig. 1, top row). Additionally, these indices merely consider water supply in drought and neglect water demand, but a drought is a condition of the water deficit between water supply and demand (Mishra and Singh, 2010). Thus, these indices do not provide sufficient information to enable decision-makers to organize a comprehensive anti-drought approach that balances all sectors of society affected by drought.

Some indices were developed with the purpose of application to all types of droughts (Fig. 1, second row). The Palmer Drought Severity Index (PDSI; Alley, 1984; Wells et al., 2004) can be applied to different types of drought by considering water supply and demand with a simplified two-layer bucket model, but it has a fixed temporal scale and does not work well over snow-covered areas (Dai, 2011a). Although the Standardized Precipitation Evapotranspiration Index (SPEI; Vicente-Serrano et al., 2010) overcomes the PDSI's weakness of fixed temporal scale, it oversimplifies complex relationships and

neglects several important hydrological processes associated with the development of drought (Zhang et al., 2015). Moreover, current indices usually use physical variables from different data sources, which inevitably introduces bias and leads to an imbalanced calculation (Naumann et al., 2014). The development of the data assimilation system brings an option of systematic input for drought indices (Xu et al., 2020). Therefore, there is a need to develop a multitype and multiscalar drought index that considers various key processes related to drought and can take full advantage of output from the data

assimilation system.

Given the abovementioned deficiency of current drought indices, Zhang et al. (2015) proposed a universal drought index, the Standardized Moisture Anomaly Index (SZI), to determine and monitor different types of droughts (Fig. 1, third row). Absorbing the strengths of the SPEI and PDSI, the SZI is available at flexible temporal scales and involves relatively sophisticated land surface processes. It builds a bridge between drought monitoring and the data assimilation system. In the

SZI, the atmospheric water demand is calculated by using variables related to evapotranspiration, runoff, soil moisture infiltration, and soil moisture loss, while water supply is taken as actual precipitation. Thus, the difference between water supply and atmospheric water demand is used to scale the water deficit and surplus. Although the SZI has been evaluated and achieved acceptable performance, it ignores the effects of snow in drought characterization, similar to the PDSI and SPEI. Such negligence in the SZI can impair its capability to monitor and identify droughts, particularly for those snow-





covered regions with considerable amounts of snowfall (Huning and Aghakouchak, 2020; Staudinger et al., 2014). To address this deficiency, Zhang et al. (2019) modified the SZI by adding snow dynamics for drought characterization into a new version of the drought index called SZI$_{snow}$ (Fig. 1, final row). This is the drought index used to construct the drought dataset in this study.

Drought is mainly characterized by severity, spatial extent, duration, and timing. The traditional method for drought
investigation is to explore the variability of its severity over a fixed study area (Hao et al., 2017). This method is also widely used to evaluate the ability of a drought index (Peng et al., 2020). Although this method can provide certain information regarding the regional drought condition, it cannot analyze the change of spatial extent with time for a drought event (Zhai et al., 2017). As drought is a spatiotemporal process, the ability of a drought index to explore the joint evolution of drought events in space and time should be given increased attention (Herrera-Estrada et al., 2017). Thus, we utilized the severity–
area–duration method (Andreadis et al., 2005; Sheffield et al., 2009), which can monitor drought in space and time, to comprehensively evaluate the drought index dataset proposed by our work.

This work aims to construct a long-term global SZI$_{snow}$ dataset for various temporal scales. The SZI$_{snow}$ is here developed to characterize multitype and multiscalar drought by accounting for different physical water–energy processes, especially snow processes. This paper is organized as follows. We first introduce the data and metrics for forcing and evaluation of the
proposed drought dataset (Sect. 2). The method behind the derivation of the SZI$_{snow}$ is summarized in Sec. 3. In Sect. 4 we present a comprehensive evaluation of the SZI$_{snow}$ to assess its ability to capture different drought types across the world, particularly over the Arctic region and Tibetan Plateau. Based on the dataset, we further analyze the spatiotemporal changes of global drought and focus on the variability of large-scale drought events. Section 5 briefly introduces how to download the proposed drought index dataset. Finally, in Sect. 6, we discuss the advancement of the SZI$_{snow}$ and its potential
applicability and implications, and present our conclusions.

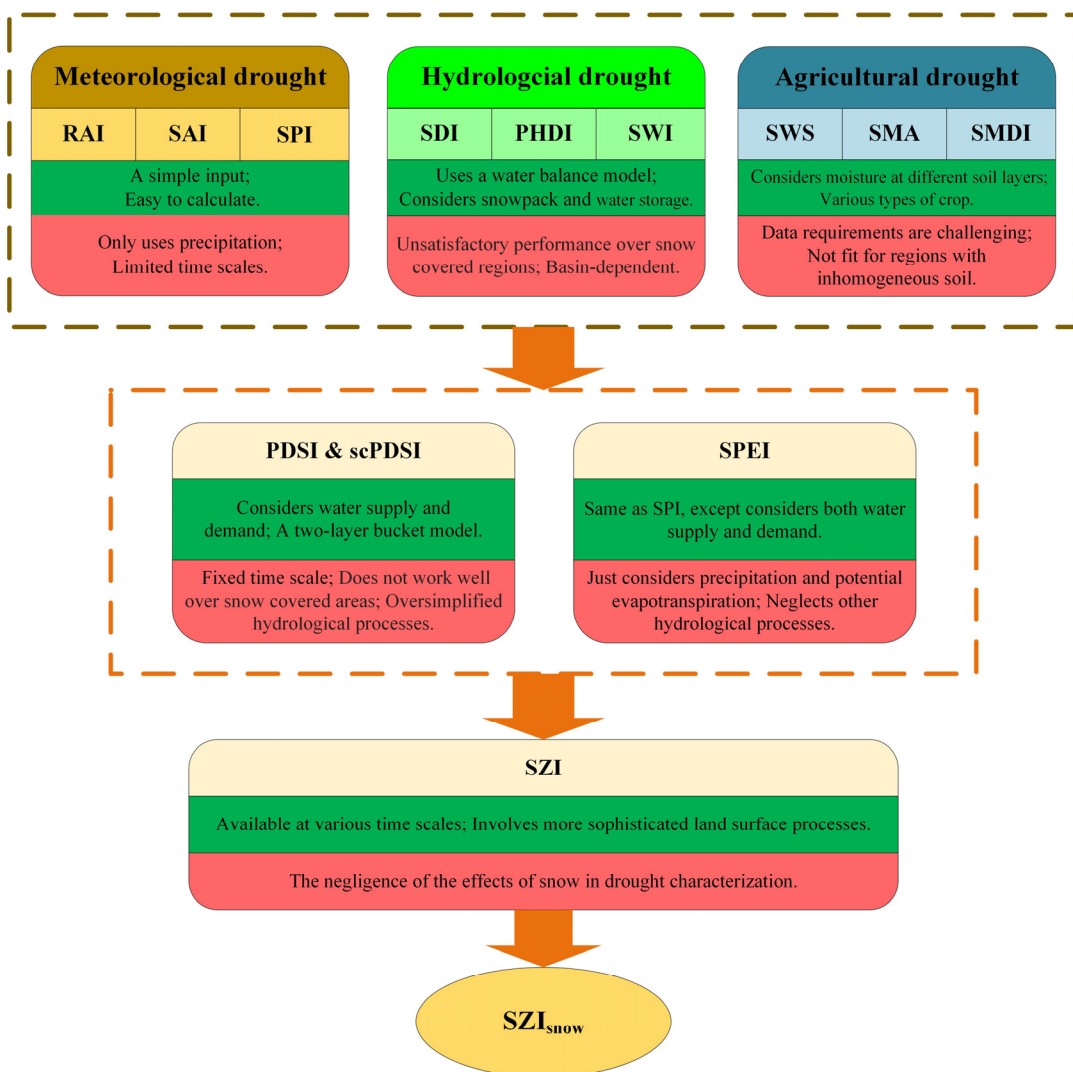

**Figure 1: Development path of the SZI$_{snow}$. Dark green boxes denote the strengths of each drought index, while pink boxes denote the weaknesses of each drought index. The top row shows indices that can only account for one type of drought, with three indices listed for each type of drought. The second row shows indices that can account for multiple types of drought. Full names of the listed indices are shown in Table S1.**

## 2 Data

### 2.1 Data for producing the SZI$_{snow}$ drought index

Hydrometeorological variables from numerical models are commonly used as the source data to compute drought indices at the global scale, due to limited observational data exist. (Sawada and Koike, 2016). Thus, in this study, the Global Land Data Assimilation System (GLDAS) provided variables to calculate the SZI$_{snow}$ globally. The SZI was also calculated for the purpose of comparison. The GLDAS is a state-of-the-art assimilation system using advanced land surface modeling and data



assimilation techniques. It incorporates satellite- and ground-based monitoring data and aims to produce optimal land surface and flux variables (Rodell et al., 2004). Currently, two versions of the GLDAS product are available: GLDAS version 1 (GLDAS-1) and GLDAS version 2 (GLDAS-2). The better performance of GLDAS-2 compared to that of GLDAS-1 has

been verified by previous work (Wang et al., 2016; Zhang et al., 2019), therefore we adopted GLDAS-2 to provide water–energy related variables to derive the $SZI_{snow}$.

The GLDAS-2 drives the Noah land surface model (LSM), forced by the global Princeton meteorological forcing data, to approximate the observed land surface state (Rui and Beaudoing, 2011). The fields of land surface states and fluxes of GLDAS-2 in this study have a 0.25° spatial resolution and monthly temporal resolution. The ability of the assimilation

system to capture the real state of the land surface is a main concern of its users. Numerous studies have assessed the meteorological forcing fields (e.g., precipitation and near-surface temperature) and modeling outputs (e.g., soil moisture and evapotranspiration) of GLDAS-2 over different regions around the world (Bi et al., 2016; Spennemann et al., 2015). The GLDAS-2 product has generally been recognized as acceptable in spite of any biases and uncertainties. Additionally, GLDAS-2 provides abundant hydrometeorological information to areas with limited observations or ungauged areas. In

particular, it bridges the gap between the scarce data available for the three poles (i.e., North Pole, South Pole, and Tibetan Plateau) and the increasing attention of the science community on these areas because of their crucial role in Earth system science.

## 2.2 Data for evaluating the performance of the $SZI_{snow}$

We firstly assessed the capability of the $SZI_{snow}$ at a basin scale across the world by using the closed terrestrial water budget

dataset developed by Pan et al. (2012). This is a monthly dataset for 32 major river basins, measured globally from 1982 to 2006. The drainage areas of these basins range from 230,000 to 600,000 km$^2$, and their locations are shown in Fig. S1. This dataset was produced based on multisource data including in situ observations, remote sensing products, land surface model simulations, and reanalysis datasets. Through a systematic assimilation strategy, the errors and biases of the multisource data were greatly compensated, which guarantees the assimilated data has the highest possible confidence. This dataset has thus

served as a baseline dataset for large basin-scale studies related to water and energy cycles and has been widely used by previous researchers (Zeng and Cai, 2016). Additionally, the variables in this dataset include precipitation, evapotranspiration, streamflow, total terrestrial water storage, and snow depth. The comprehensive variables in the dataset facilitate the calculation of different drought indices as references to evaluate the $SZI_{snow}$.

We applied a drought index, the Standardized Wetness Index (SWI), to evaluate the performance of the SZI and $SZI_{snow}$ at

the global scale. The details of the SWI will be introduced in Sect. 2.3. The datasets used to calculate the SWI include the Climatic Research Unit Time Series (CRU TS) Version 4.01 and the Global Land Evaporation Amsterdam Model (GLEAM) Version 3.1a. The CRU TS supplies monthly precipitation (P) and potential evapotranspiration (PET) data at a spatial resolution of 0.5° (https://crudata.uea.ac.uk/cru/data/hrg/). This PET data is computed by the Penman–Monteith equation. The GLEAM provides monthly actual evapotranspiration (ET) data at a spatial resolution of 0.25° (https://www.gleam.eu/).





Both the CRU TS and GLEAM cover a period from 1980 to 2010. In addition, the GLEAM dataset was interpolated from
the spatial resolution of 0.25° to that of 0.5° to facilitate the computation of the SWI.

**2.3 Evidence of different drought types for the SZI$_{snow}$ evaluation**

We evaluated the ability of the SZI$_{snow}$ and SZI to capture different types of droughts based on drought evidence. First,
meteorological, hydrological, and agricultural drought evidence was identified based on precipitation, streamflow, and soil
water storage, respectively, from the dataset of Pan et al. (2012) (as mentioned in Sect. 2.2) over the 32 large basins. Then,
the evidence was compared with the SZI$_{snow}$ and SZI, calculated based on the GLDAS-2 product. In addition, for the
convenience of comparison, we adopted the log–logistic distribution to standardize precipitation, streamflow, and soil water
storage for the computation of the Standardized Precipitation Index (SPI; Mckee et al., 1993), Standardized Streamflow
Index (SSI; Vicente-Serrano et al., 2012) and Standardized Water Storage Index (SWSI; Aghakouchak, 2014).

We also selected the residual water–energy ratio (WER) as a comprehensive drought indicator to evaluate the SZI$_{snow}$. The
WER is defined as the ratio of residual available water (P–ET) to residual energy (PET–ET), and can integrally reflect
drought conditions by depicting variation in water–energy balance. The WER was first suggested by Liu et al. (2017) as
many studies found that the ratio of sensible heat (the residual energy supply after dissipating through latent heat) to net
radiation (total energy supply) is always raised under drought, meanwhile the ratio of residual available water to
precipitation (total water supply) is always lowered under drought. Consequently, the WER is lowered during drought and
can be used as a comprehensive drought indicator. Again, we used independent datasets (i.e., the CRU TS and GLEAM
datasets) to globally calculate the WER and compare it with the SZI$_{snow}$ and SZI. As for the SSI and SWSI, the WER was
standardized for the calculation of the SWI.

**2.4 Metrics for the SZI$_{snow}$ evaluation**

This study applied the SPI (a meteorological drought index), SSI (a hydrological drought index), SWSI (an agricultural
drought index), and SWI (a comprehensive drought index) as references to evaluate the performance of the SZI$_{snow}$ and SZI.
The four referenced drought indices were computed with datasets that were independent from the dataset used for calculating
the SZI$_{snow}$ and SZI. We utilized Pearson correlation coefficients (r) of SPI–SZI/SZI$_{snow}$, SSI–SZI/SZI$_{snow}$, SWSI–SZI/SZI$_{snow}$,
and SWI–SZI/SZI$_{snow}$ to compare the performance of the SZI and SZI$_{snow}$ in terms of their capacity to capture multitype and
multiscalar drought across different climate zones.





# 3 Methodology

## 3.1 Derivation of the SZI$_{snow}$

### 3.1.1 Hydrologic accounting

The physical processes included in the construction of the SZI$_{snow}$ are shown in Fig. 2. Six water budget components are
involved in the procedure of hydrological accounting to determine the water demand over a region. The related variables
comprise ET, PET, runoff, potential runoff, soil infiltration, potential soil infiltration, soil moisture loss, potential soil
moisture loss, snow water equivalent (SWE) accumulation, potential SWE accumulation, snowmelt, and potential snowmelt.
The monthly values of these variables were derived from land surface models, for instance, the GLDAS-2 Noah LSM in the
present study. The prominent improvement of the SZI$_{snow}$ is that it accounts for the influence of snowfall on hydrological
processes, which was completely ignored in the SZI (Zhang et al., 2019; Zhang et al., 2015).

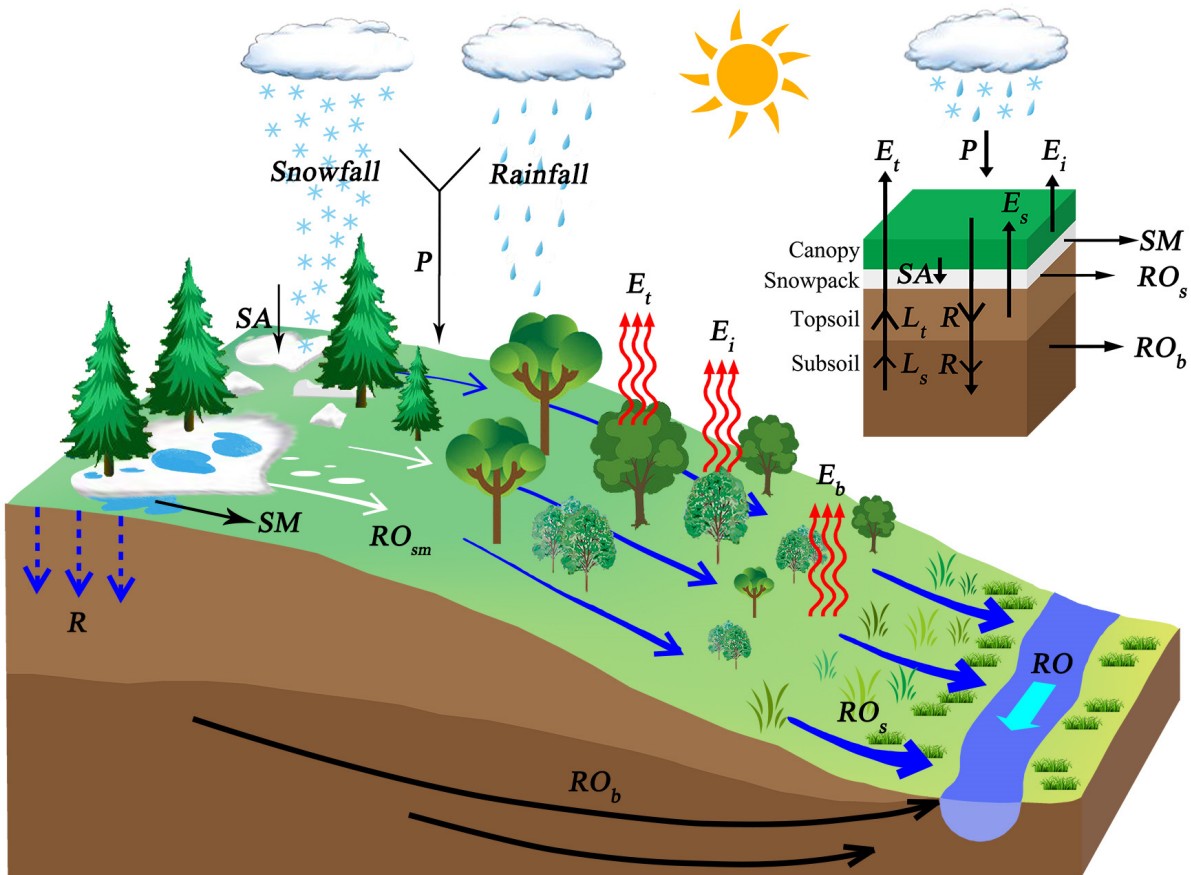

**Figure 2:** **Schematic diagram of the relative physical processes included in the construction of the SZI$_{snow}$. The upper right subplot
shows the variables used in the SZI$_{snow}$ calculations at the pixel level. Full names of all the abbreviations in this figure are listed in**
**Table S2.**



Both the soil moisture storage and snow storage are considered as reservoirs in the SZI$_{snow}$, which is different from the SZI that solely considered the former. Changes in soil moisture storage (soil infiltration or soil moisture loss) and snow (SWE accumulation or snowmelt) can alter the regional water balance (water supply or water demand), and then affect the drought condition. Consequently, the SZI$_{snow}$ contains more comprehensive hydrological processes than the SZI. The improvement of

the SZI$_{snow}$ makes it applicable to a wider variety of climatic regions, especially for regions that belong to the three poles where more snow is stored than at any other place on Earth. Detailed equations for the hydrologic accounting in the SZI$_{snow}$ are listed in Table 1. All full names for abbreviations contained in the equations are supplied in Table S2.

**Table 1. The procedures, variables, and associated equations used to calculate the SZIsnow.**

| Procedures | Variables | Equations | |
|---|---|---|---|
| Hydrological accounting | R/PR | $R = \begin{cases} \Delta S_t + \Delta S_u & \Delta S_t + \Delta S_u \geq 0 \\ 0 & \Delta S_t + \Delta S_u < 0 \end{cases}$ <br> $PR = AWC - (S_t + S_u)$ | (1) |
| | RO/PRO | $RO = RO_s + RO_b + RO_{sm}$ <br> $PRO = AWC - PR$ | (2) |
| | ET/PET | $ET = E_b + E_t + E_i$ <br> $PET$ is computed with Penman-Monteith equation | (3) |
| | L/PL | $\begin{cases} L = \begin{cases} 0 & \Delta S_t + \Delta S_u \geq 0 \\ -(\Delta S_t + \Delta S_u) & \Delta S_t + \Delta S_u < 0 \end{cases} \\ PL = PL_t + PL_s \\ \quad \begin{cases} PL_t = Min(PET, S_t) \\ PL_s = (PET - PL_t)\dfrac{S_u}{AWC} \end{cases} \end{cases}$ | (4) |
| | SA/PSA | $\begin{cases} SA = \begin{cases} 0 & \Delta SWE < 0 \\ \Delta SWE & \Delta SWE \geq 0 \end{cases} \\ PSA = P_{snow} \end{cases}$ | (5) |
| | SM/PSM | $\begin{cases} SM = \begin{cases} -\Delta SWE & \Delta SWE < 0 \\ 0 & \Delta SWE \geq 0 \end{cases} \\ PSM = SWE \end{cases}$ | (6) |
| Climatic coefficients | | $\alpha_j = \overline{ET_j}/\overline{PET_j}$ <br> $\beta_j = \overline{R_j}/\overline{PR_j}$ <br> $\gamma_j = \overline{RO_j}/\overline{PRO_j}$ <br> $\delta_j = \overline{L_j}/\overline{PL_j}$ <br> $\varepsilon_j = \overline{SA_j}/\overline{PSA_j}$ <br> $\varphi_j = \overline{SM_j}/\overline{PSM_j}$ | (7) |
| CAFEC | | $\hat{P}_{snow} = \alpha_j PET + \beta_j PR + \gamma_j PRO + \delta_j PSA - \varepsilon_j PL - \varphi_j PSM$ | (8) |
| Standardization | | $\begin{cases} P = P_{rainfall} + P_{snowfall} \\ Z_{snow} = P - \hat{P}_{snow} \end{cases}$ | (9) |



The regional water supply firstly meets water demand from soil layers. The soil infiltration (R) was estimated by monthly changes ($\Delta S_t$ and $\Delta S_u$) in available soil moisture of the top ($S_t$) and bottom ($S_u$) soil layers. The potential soil infiltration (PR) was calculated as the difference between the available soil water capacity (AWC) and the available soil moisture of the entire soil. AWC is estimated as the maximum soil water of the two soil layers (Fig. 2) in the Noah LSM. Then, the rest of the regional water supply satisfies water demand from runoff (RO). RO consists of surface runoff ($RO_s$), baseflow ($RO_b$), and snowmelt runoff ($RO_{sm}$), which are directly obtained from the GLDAS-2. The potential runoff (PRO) is the difference between AWC and PR, because soil moisture storage is considered as a water reservoir in the $SZI_{snow}$. Additionally, water supply is partly consumed by ET, including bare soil evaporation ($E_b$), transpiration ($E_t$), and canopy water evaporation ($E_i$) that can be found in the output of the GLDAS-2. The PET is computed with output fields from GLDAS-2 using the Penman–Monteith equation. Moreover, the moisture loss (L) from the soil layers is considered in the $SZI_{snow}$. The equations for L and its potential values are shown in Equation 4 of Table 1. Lastly, calculations of variables related to snow processes underscored by the $SZI_{snow}$ are given in Equations 5–6 of Table 1. The potential snow accumulation (PSA) equals the monthly amount of snowfall ($P_{snow}$), and the monthly SWE change completely reflects snow accumulation (SA) and snowmelt (SM).

### 3.1.2 Climatic coefficients and precipitation that is climatically appropriate for existing conditions (CAFEC)

Similar to the PDSI, the $SZI_{snow}$ applies the precipitation amount that is climatically appropriate for existing conditions (CAFEC, referred as $\widehat{P}_{snow}$) to quantify the regional water demand. The amount of $\widehat{P}_{snow}$ is the result of interaction among the six water budget components as shown in Equation 8 of Table 1. The weighting factor for each component is the climatic coefficient, which is defined as the ratio of the monthly climatic averages of actual (water supply) to potential (water demand) values. The equations used to compute these climatic coefficients are listed in Equation 7 of Table 1. The j in Equation 7 denotes months of a year, that is, each water budget component has 12 values of climatic coefficient covering all months.

### 3.1.3 Standardization of moisture anomaly

The comparison between the actual precipitation (P) and $\widehat{P}_{snow}$ can reflect the drought condition. When the actual P is less than $\widehat{P}_{snow}$, the regional water supply will remain in deficit, and vice versa for a surplus. Thus, the difference between the actual P and $\widehat{P}_{snow}$ is an appropriate indicator for regional water deficiency or surplus. Such difference is defined as the moisture anomaly $Z_{snow}$ (Equation 9 of Table 1) in the drought assessment system of the $SZI_{snow}$. In addition, the $Z_{snow}$ can be aggregated at different temporal scales (e.g., 1–48 months) with the same processes as for the SPEI. For a detailed procedure, readers should refer to the paper of Vicente-Serrano et al. (2010). Moreover, we standardized the $Z_{snow}$ to $SZI_{snow}$ to realize the comparability of the $Z_{snow}$ with other $Z_{snow}$ or other drought indices over space and time. A three-parameter log–logistic distribution was adopted to standardize $Z_{snow}$ time-series and derive the $SZI_{snow}$. This follows the same approach used for the SPEI and SWI to standardize $Z_{snow}$ at different temporal scales. The average value of the

$SZI_{snow}$ is 0, and the standard deviation is 1. Finally, we scaled the $SZI_{snow}$ categorization levels to the corresponding SPEI
drought severity categories in Table 2 because the same standardization method is used for both.

**Table 2. Categorization of wetness and drought conditions in the SZI$_{snow}$.**

| Categories | SZI$_{snow}$ values |
|---|:---:|
| Wetness categorization | |
| Extreme wetness | ≥2.0 |
| Severe wetness | 1.5–2.0 |
| Moderate wetness | 1.0–1.5 |
| Mild wetness | 0.5–1.0 |
| Near normal | –0.5 to 0.5 |
| Drought categorization | |
| Mild drought | –1.0 to –0.5 |
| Moderate drought | –1.5 to –1.0 |
| Severe drought | –2.0 to –1.5 |
| Extreme drought | <–2.0 |

## 3.2 Identification of large-scale drought events in space and time

Using the $SZI_{snow}$ dataset constructed by this study, we performed a global and continental drought analysis for the period
1948–2010. We focused on the temporal variability of large-scale drought events through a severity–area–duration (SAD)
drought diagnosis method (Andreadis et al., 2005; Herrera-Estrada et al., 2017). In contrast to traditional studies which
analyze the intensity, severity, and duration of drought over a fixed region, the SAD method specializes in simultaneously
tracking the development of droughts in space and time based on a gridded dataset. This method proposes a Lagrangian
approach by aggregating grids (under specified drought levels) of contiguous areas into clusters. These clusters are then
tracked and archived as they propagate through space and time. The main steps of the SAD method are outlined as follows.
The SAD method firstly uses a monthly three-dimensional (3D) gridded drought index dataset to identify two-dimensional
(2D) drought clusters in each time step. This drought cluster identification procedure is built on a clustering algorithm that
merges spatial contiguity. Then, a median filter is utilized to smooth out noise (i.e., small-scale heterogeneity) in the 2D
clusters in each time step. Specifically, we regarded a grid with a $SZI_{snow}$ value below -1.0 as being under drought and
considered connected areas within which all the grids had a $SZI_{snow}$ below -1.0 as a drought cluster. The last step of the
clustering procedure is to remove clusters with an area less than 500,000 km$^2$. The remaining clusters are regarded as
separate drought events for each time step. More importantly, these identified drought clusters are allowed to split or merge
through time in the SAD method. The tracking algorithm links clusters that have overlapping grid cells and records the
merging or splitting date, areas, and centroids of clusters. These functions in the SAD method make it possible for us to
monitor the spatiotemporal evolution of large-scale drought events. A large-scale drought event in North America identified
by the SAD method is given as an example in Fig. S2 to illustrate this method.





## 4 Results

### 4.1 Evaluation of the SZI$_{snow}$

#### 4.1.1 Evaluation of the SZI$_{snow}$ for different drought types

We firstly evaluated the capability of the SZI$_{snow}$ to capture meteorological drought at different temporal scales over the 32 large basins from 1948 to 2010, and compared this to the capability of the SZI. The observation-based meteorological drought index, SPI, was used as a reference. As shown in Fig. 3a, the blue boxes represent the statistical distribution of the Pearson correlation coefficient (r) between the SZI and SPI for 1–48 month timescales in each basin, while the red boxes represent that between the SZI$_{snow}$ and SPI. The SZI$_{snow}$ generally outperformed the SZI over these large basins with an

average improvement of 3.19% (ranging from 0.01% to 6.45%). It is clear that the extent of improvement in the SZI$_{snow}$ increases as the SWE increases. For instance, the r of the SZI$_{snow}$–SPI is 5.51% higher than that of the SZI–SPI in the Pechora basin that has an SWE of 47.5 mm yr$^{-1}$. In contrast, the r of the SZI$_{snow}$–SPI is only 0.11% higher than that of the SZI–SPI in the Indus basin that has an SWE of 3.70 mm yr$^{-1}$. The relationship between the enhancement in the SZI$_{snow}$ and SWE implies that the SZI$_{snow}$ brings advantages in accounting for snow processes. It also demonstrates that the SZI$_{snow}$

appropriately reflects the fact that snow accumulation and melt have considerable impacts on the seasonal and inter-annual variation of streamflow in snow-covered areas. In summary, the SZI$_{snow}$ has a satisfactory performance to capture meteorological drought.

We then evaluated the capability of the SZI$_{snow}$ to capture hydrological drought (Fig. 3b). The observation-based hydrological drought index, SSI, was used as a reference. The SZI$_{snow}$ generally outperformed the SZI over these large basins,

with an average improvement of 3.13% (ranging from 0.25% to 17.53%). The extent of improvement in the SZI$_{snow}$ increases as the SWE increases. For instance, the r of the SZI$_{snow}$–SSI is 17.53% higher than that of the SZI–SSI in the Pechora basin that has an SWE of 47.5 mm yr$^{-1}$. In contrast, the r of the SZI$_{snow}$–SSI is only 1.18% higher than that of the SZI–SSI in the Indus basin that has an SWE of 3.70 mm yr$^{-1}$. Moreover, among the multiple temporal scales over which it was tested, the SZI$_{snow}$ performs best at the 12-month scale for hydrological droughts. At the 12-month scale, the SZI$_{snow}$ performs 17.53%,

11.46%, 19.40%, and 4.88% better than the SZI in the Pechora, Northern Dvina, Yenisei, and Kolyma basins, respectively. Thus, the SZI$_{snow}$ performs well in the context of capturing hydrological drought.

The capability of the SZI$_{snow}$ and SZI to capture agricultural drought was also assessed in our study. We conducted the same steps of assessment as those for assessing hydrological drought, but the reference drought index was an agricultural drought index, SWSI, derived from a dataset based on observations. As shown in Fig. 3c, the average r of the SZI$_{snow}$–SWSI for all

the basins is 0.51 (ranging from 0.19 to 0.79), which indicates the ability of the SZI$_{snow}$ to reliably capture agricultural drought. Additionally, the SZI$_{snow}$ performs better than the SZI in almost all basins, and the average improvement of the SZI$_{snow}$ is 6.46% (ranging from 0.14% to 38.96%). Again, larger improvements occurred in basins with a larger SWE. This comparison, in terms of agricultural drought, again emphasizes the strength of the SZI$_{snow}$ in high-latitude and high-altitude regions with relatively greater SWE. In summary, the SZI$_{snow}$ performs sufficiently well to capture agriculture drought.

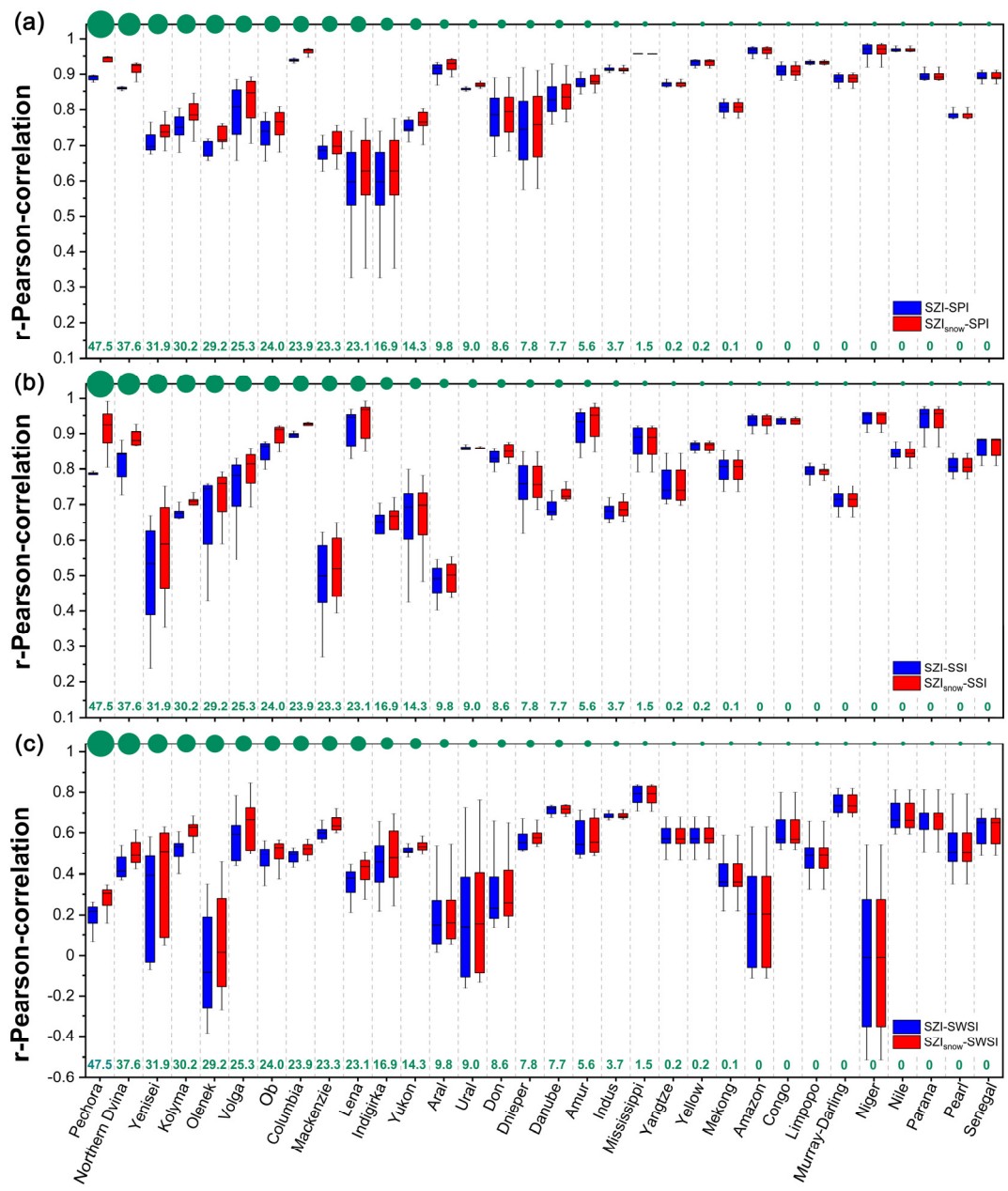


**Figure 3: Comparisons between the SZI$_{snow}$ and SZI with regard to their performance in quantifying different types of drought. The SZI$_{snow}$ and SZI were correlated with observed drought indices across the 32 basins. The Pearson correlation coefficient (r) was applied to evaluate the correlation. (a) Performance of the SZI$_{snow}$ in quantifying meteorological drought. The blue boxes represent the statistical distribution of r between the SZI and SPI for timescales from 1 to 48 months in each basin, while the red**
**boxes represent that between the SZI$_{snow}$ and SPI. (b) Performance of the SZI$_{snow}$ in quantifying hydrological drought (boxes represent same parameters as in (a) but correlations are with the SSI instead of the SPI). (c) Performance of the SZI$_{snow}$ in quantifying agricultural drought (boxes represent same parameters as in (a) but correlations are with the SWSI instead of the SPI). The SPI, SSI, and SWSI were derived by standardizing observed precipitation, streamflow, and soil water storage data. Basins were ranked in descending order based on their SWE. The green dots along the top of each x-axis denote the SWE of each basin**
**and their sizes were scaled by the green SWE values given along the bottom of each x-axis.**



### 4.1.2 Evaluation of the SZI$_{snow}$ across different spatial scales

The SZI$_{snow}$ can be computed and used to characterize drought for an individual grid, although we evaluated its capability at a basin scale in Sect. 4.1.1. In this section we apply the SWI drought index as a reference to assess the SZI$_{snow}$ at the global scale (Figs. 4a–4b). The Hovmöller diagram (Fig. 4a) shows the distribution of the difference between the r of the
SZI$_{snow}$– SWI and that of the SZI–SWI for 1–48 month timescales across different latitudes. It is clear that the high-value zonal mean difference mainly centers in the interval of 50–65° N. This indicates that the SZI$_{snow}$ outperformed the SZI within this 15° interval in high latitude areas. In contrast, the remaining regions, outside of this interval, show only small magnitude differences. In addition, as shown in Fig. 4b, the improvement of the SZI$_{snow}$ varies over different timescales; it performs better over timescales in the range of 3–12 months. Such spatial patterns, as shown in Figs. 4a–4b, emphasize the physical
improvement in terms of snow processes in the SZI$_{snow}$ construction compared to the SZI. This evaluation shows the appropriate performance of the SZI$_{snow}$ at the global scale.

As the three-pole region is a focus of this study, we specifically compared the SZI$_{snow}$ and SZI over the Arctic region, where the latitude is larger than 66° 33' N. Figs. S3a–S3b presents the spatial distributions of the r of the SZI$_{snow}$–SWI and SZI– SWI, respectively, over a 12-month timescale. The two maps show similar spatial patterns for the SZI$_{snow}$ and SZI, yet
the r of the SZI$_{snow}$–SWI is larger than that of the SZI–SWI over the majority of the Arctic region, indicated by the positive difference shown in Fig. 4c. Once again, the SZI$_{snow}$ is seen to outperform the SZI over the Arctic region, which is consistent with the findings from the global evaluation shown in Figs. 4a–4b. Additionally, the relationship of area-averaged r and timescales is shown in Fig. 4d. The maximum r appears when the timescale is 12-months, and the relative difference between the r of the SZI$_{snow}$–SWI and that of the SZI–SW (i.e., the improvement of the SZI$_{snow}$) shows a rapid growth
moving from 1- to 12-month timescales (Fig. 4d, insert plot). The results demonstrate that the SZI$_{snow}$ dataset performs well over the Arctic region.

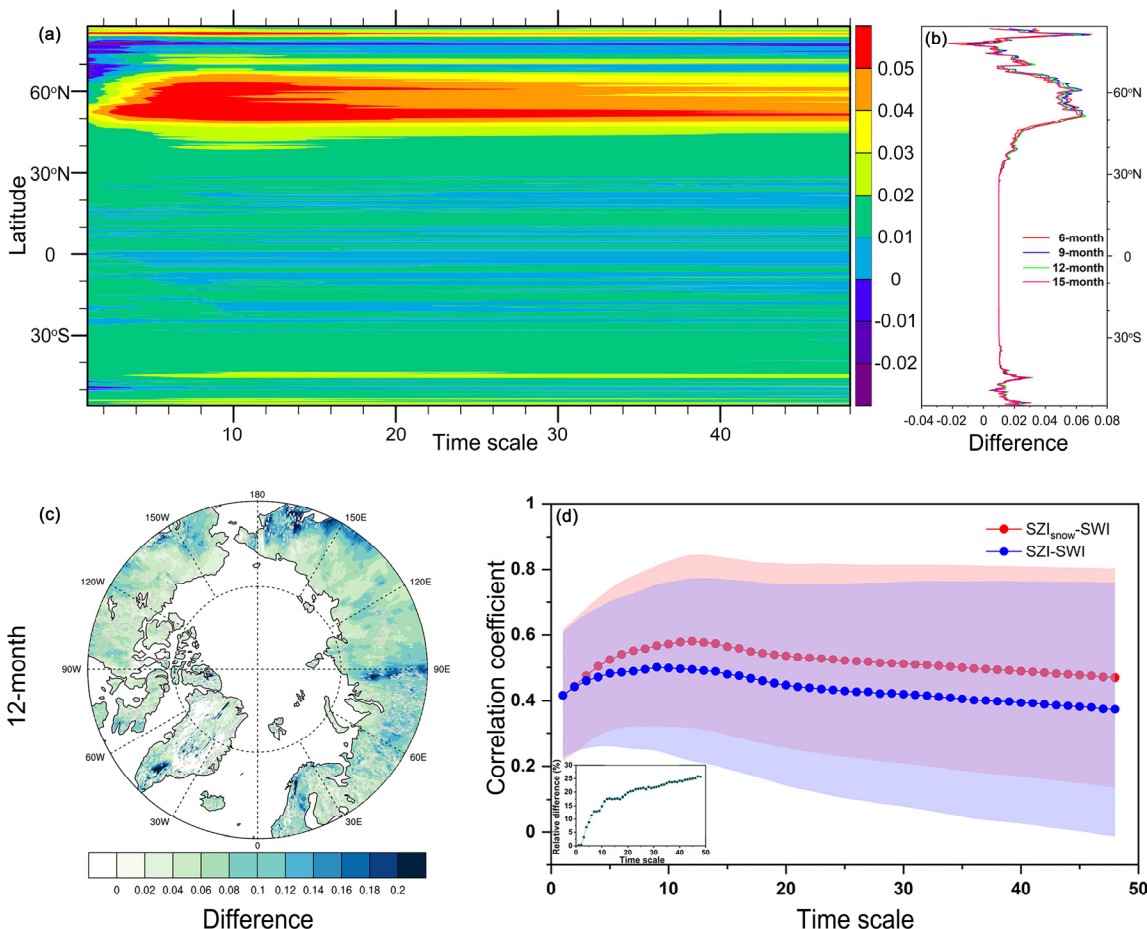

**Figure 4: Performance of the SZI$_{snow}$ over different latitudes (a and b) and specifically over the Arctic region (c and d). Here the differences between the correlation coefficients of the SZI$_{snow}$–SWI and those of the SZI–SWI for different timescales were used to**
**compare their performance. (a) The Hovmöller diagram (timescale × latitude) shows the differences averaged by latitude from 55°S to 85°N for timescales ranging from 1 to 48 months. (b) Distribution of the difference for specific timescales (6, 9, 12, and 15 months) with changing latitude. (c) Spatial distribution of the differences between the correlation coefficients of the SZI$_{snow}$–SWI and those of the SZI–SWI over a 12-month timescale in the Arctic region. (d) Variations of correlation coefficients averaged over the Arctic region for various temporal scales. The inset shows the change of relative difference (%) for these temporal scales.**

The risk of drought on the Tibetan Plateau, the world's third pole, can affect the water supplies of billions of people. Figure 5 shows the capability of the SZI$_{snow}$ and SZI to capture drought at various temporal scales over the Tibetan Plateau. Both the SZI$_{snow}$ and SZI have high r values with the SWI over a large part of the Tibetan Plateau. The r of the SZI$_{snow}$–SWI is larger than 0.6 across 68.96% of the entire Tibetan Plateau, and for the SZI–SWI this value is 61.93% (Fig. 5, left and central columns). The area-averaged r of the SZI$_{snow}$–SWI is 0.72 and that of the SZI–SWI is 0.65 over a 12-month timescale,
equating to an improvement of 10.77% for the SZIsnow. Moreover, the phenomenon that the SZI$_{snow}$ outperforms the SZI is clearly shown in the right column of Fig. 5. The largest improvement is seen mainly in the northwest corner and southeastern part of the Tibetan Plateau, where the largest snow depths are also seen (Dai et al., 2017). Thus, the SZI$_{snow}$ dataset is a reliable resource to quantify drought across the Tibetan Plateau.

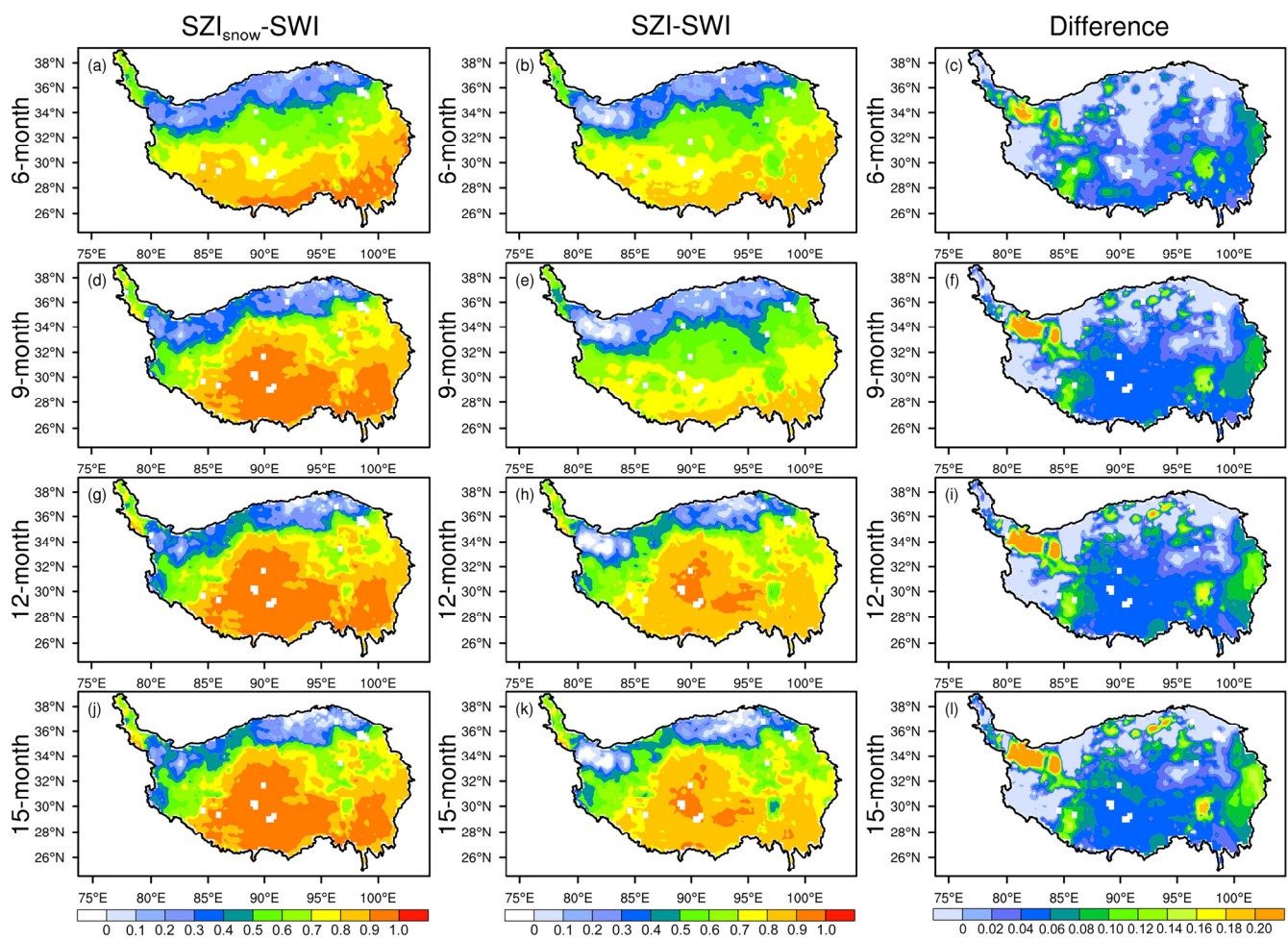

Figure 5: Spatial distribution of correlation coefficients of the SZI$_{snow}$–SWI (left column) and those of the SZI–SWI (middle column), and the differences between the two (right column = left column minus middle column) over the Tibetan Plateau at different timescales (6, 9, 12, and 15 months).

## 4.2 Historical trends in global drought

The proposed SZI$_{snow}$ dataset was applied to investigate the historical changes in global drought between 1948 and 2010. The spatial distribution of the linear trend in the SZI$_{snow}$ over different timescales (i.e., 3-, 6-, 12-, 15-months) is shown in Fig. 6. The SZI$_{snow}$ at each temporal scale demonstrates a similar global pattern, except for differences in the magnitude of dryness or wetness trends. Overall, 59.66% of the land area of the Earth displays a drying trend, and the remaining 40.34% exhibits a wetting trend. As shown in Fig. 6, the SZI$_{snow}$ shows a drying trend over eastern Asia, northern India, most of the Arabian Peninsula and Africa, eastern Australia, and central and southern Europe; increased wetness was found over most of the United States, a large part of South America, and central Australia. Additionally, the drying trend tends to increase as the timescale becomes longer. For instance, the drying rate of the SZI$_{snow}$ over eastern Asia becomes larger as its timescale



increases. Moreover, our results are broadly consistent with the findings of Dai (2013) who analyzed the trend of global drought using the self-calibrated PDSI. This also implies that the $SZI_{snow}$ is a useful proxy of aridity changes.

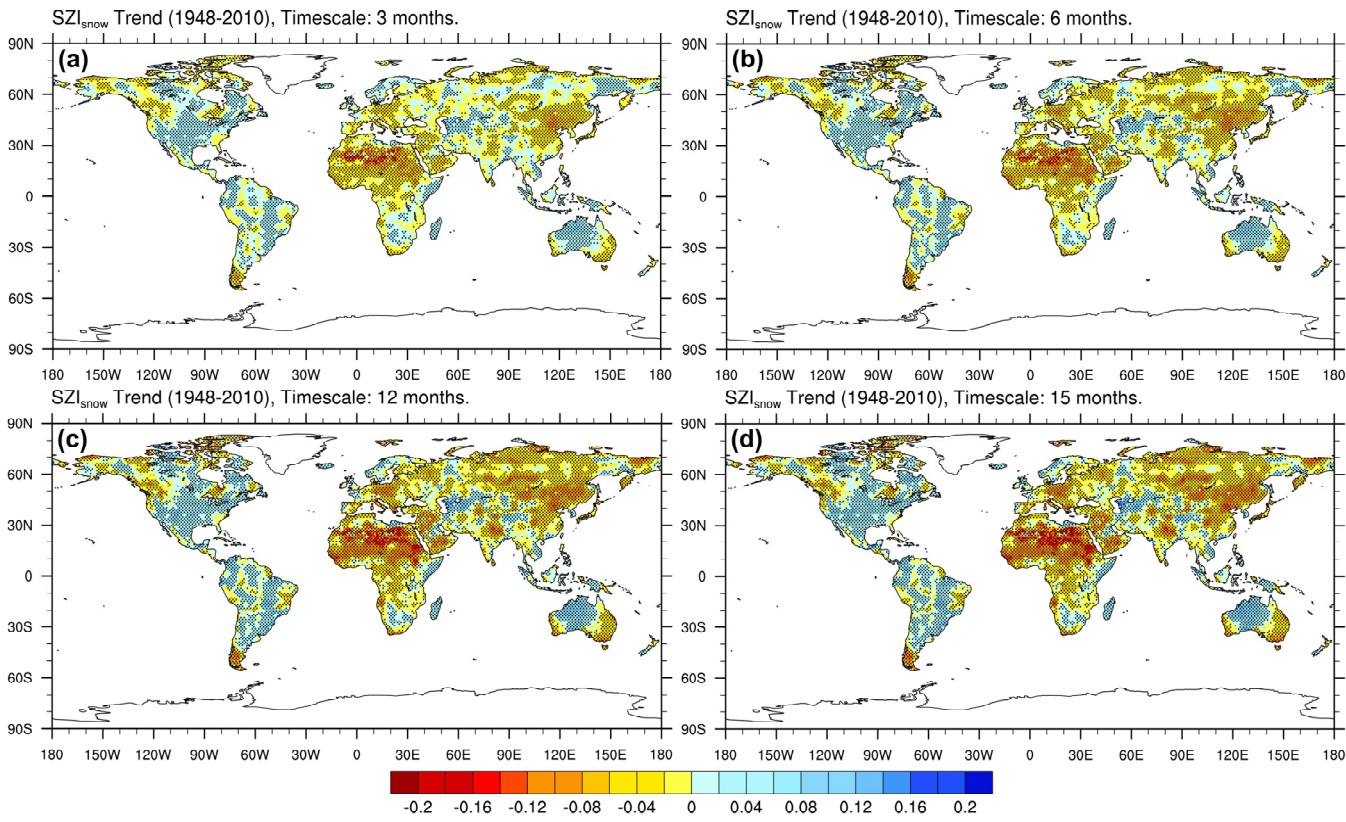

**Figure 6: Spatial distribution of the linear trend (changes per 50 years) in the $SZI_{snow}$ during the period 1948–2010, at various timescales. The stippling denotes the trend being statistically significant at the 95% confidence level.**

We further examined variations in the area of global land under drought (Fig. 7a). The area under drought shows an increasing trend with an average rate of increase of 0.05% $yr^{-1}$. Large fluctuations began to emerge from 1975, and Earth's drought area increased rapidly in the early 1980s. This growth was largely attributed to the leap in temperature caused by the 1982–1983 El Niño (Timmermann et al., 1999; Dai, 2011b). The maximum extent of drought area appeared in 1991. Moreover, the temporal change in the global moisture anomaly $Z_{snow}$ is shown in Fig. 7b. The $Z_{snow}$ displays a global downward trend of −0.11 mm $yr^{-1}$ for the period 1948–2010, which indicates the increasing global deficit between water supply and water demand. Overall, our analysis based on the $SZI_{snow}$ dataset revealed increased aridity over many land areas, and severe and widespread droughts over the Earth since 1948.

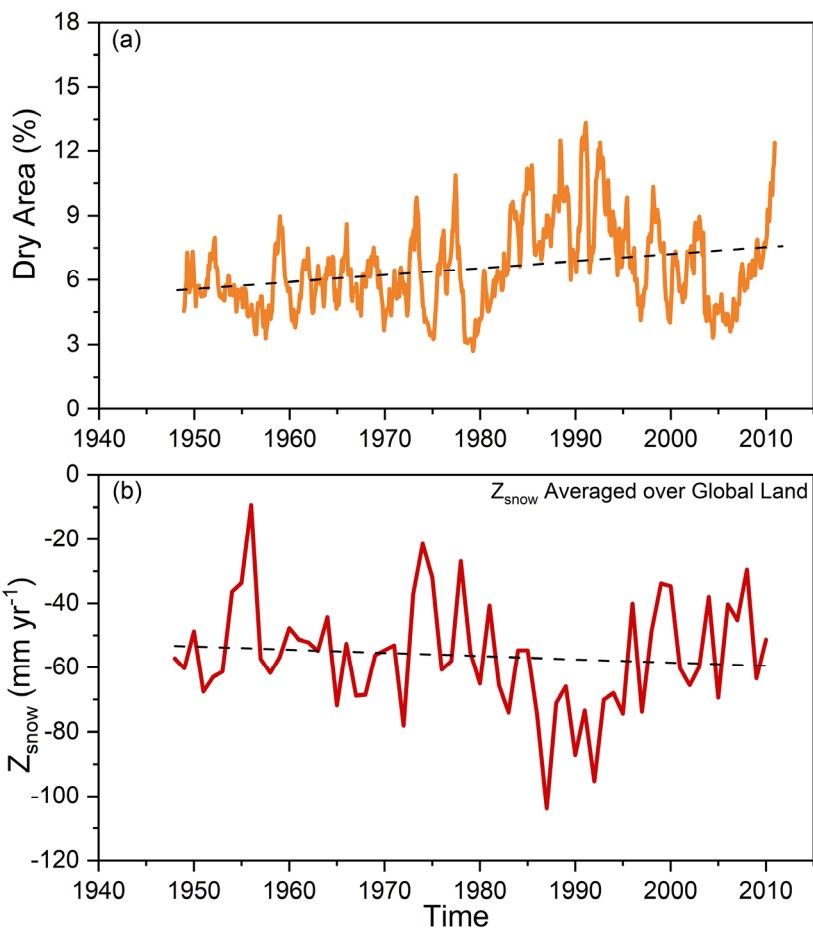

**Figure 7: Time series of (a) global dry land area (% yr⁻¹) and (b) $Z_{snow}$ (mm yr⁻¹) between 1948 and 2010. The dry land area was calculated based on the $SZI_{snow}$ at a 12-month timescale. The dashed lines denote the linear trends.**

### 4.3 Global and continental large-scale drought events

**4.3.1 Statistics of large-scale drought events**

Using the $SZI_{snow}$ dataset proposed in this study, we analyzed global and continental large-scale drought events (hereinafter referred to as drought) from 1948 to 2010 by leveraging the SAD drought diagnosis method. There have been 525 droughts with an area larger than 500,000 km² globally during the study period, as shown in Table 3. Also outlined in Table 3 is detailed information for the droughts with the longest duration and the largest area, respectively, for each continent.

Droughts with a duration longer than 6 months account for nearly 70% of all droughts. The longest drought that occurred in North America lasted 37 months from 1964 to 1967. The most spatially extensive drought occurred over Asia in August 2008 (drought lasted from November 2007 to June 2009) and covered an area of approximately 11 million km² (roughly 100 times the size of Guatemala). For comparison, the most extensive drought in Oceania covered nearly 66% of its continental area (roughly 54 times the size of Guatemala).





**Table 3. Summary of large-scale drought occurrence for each continent. In the fourth column, the duration of the drought is shown in months, and the period is listed in parentheses. In the final column, the spatial extent given as a percentage of the total continental area, and the date at which the maximum spatial extent occurred, is listed in parentheses.**

| Region | Number of droughts | Number of droughts ≥ 6 months | Longest duration (months) | Maximum spatial extent (km$^2$) | |
|---|---|---|---|---|---|
| Asia | 168 | 108 | 28 (1983-86) | 10 828 203 | (24.3%, August 2008) |
| Europe | 30 | 19 | 11 (2005) | 3 667 854 | (36.1%, Sept 1992) |
| Africa | 98 | 66 | 27 (1993-95) | 9 953 960 | (32.9%, August 1984) |
| Oceania | 39 | 30 | 21 (1976-78) | 5 897 639 | (65.7%, Sept 1994) |
| North America | 104 | 71 | 37 (1964-67) | 7 339 018 | (30.3%, April 2000) |
| South America | 86 | 65 | 29 (1957-59) | 9 510 882 | (53.3%, October 1963) |

We further ranked the top five droughts in terms of duration and maximum spatial extent for each continent (Table 4). For
Asia, the longest drought lasted 28 months, and its droughts commonly extend across larger areas compared to other continents. The top five longest droughts in Europe had a relatively short duration compared to other continents. In Africa, the longest drought lasted 27 months, and the maximum extent was 10 million km$^2$; of all analyzed droughts, 60% occurred in the period from the mid-1980s to the mid-1990s; it is clear that there was a prolonged drought spell over this period. Moreover, the droughts in North America always have a longer duration compared to other continents.



**365** **Table 4. Top five drought events in each continent, ranked by duration, or by maximum spatial extent. The duration and spatial extent are listed in parentheses after the period of each drought event.**

| Region | Duration (months) | Spatial extent ($10^6$ km$^2$) | |
|---|---|---|---|
| Asia | 1983-86 (28) | 2007-09 | (10.8) |
| | 2005-07 (27) | 1988-89 | (10.8) |
| | 2007-09 (20) | 1990-91 | (8.8) |
| | 1996-98 (20) | 1972-73 | (8.4) |
| | 1992-94 (20) | 1996-98 | (8.0) |
| Europe | 2005 (11) | 1992-93 | (3.7) |
| | 1992-93 (11) | 1990-91 | (3.3) |
| | 1990-91 (11) | 1993 | (3.2) |
| | 2009-10 (9) | 2005-06 | (3.1) |
| | 1993 (9) | 1973 | (2.5) |
| Africa | 1993-95 (27) | 1984-85 | (10.0) |
| | 1980-82 (25) | 1982-84 | (9.6) |
| | 1991-93 (22) | 1987-88 | (9.6) |
| | 1989-91 (19) | 1991-92 | (6.8) |
| | 1985-87 (19) | 1982-83 | (6.4) |
| Oceania | 1976-78 (21) | 1994-95 | (5.9) |
| | 1951-53 (21) | 1964-65 | (5.4) |
| | 2006-07 (16) | 1961-62 | (5.1) |
| | 1961-62 (14) | 1951-53 | (5.0) |
| | 1972-73 (13) | 1972-73 | (5.0) |
| North America | 1964-67 (37) | 1998-00 | (7.3) |
| | 1959-62 (27) | 1976-77 | (6.6) |
| | 1979-82(26) | 1962-64 | (6.6) |
| | 2001-02 (24) | 1952-53 | (6.3) |
| | 1998-00 (24) | 1979-82 | (6.1) |
| South America | 1957-59 (29) | 1963-64 | (9.5) |
| | 1960-62 (25) | 1997-98 | (7.0) |
| | 1995-96 (24) | 1988-89 | (6.5) |
| | 1991-93 (23) | 1991-93 | (5.8) |
| | 2008-09 (20) | 1957-59 | (5.2) |





### 4.3.2 Temporal variability of large-scale drought events

The temporal variation of area-averaged $SZI_{snow}$, the area under drought (pixels with $SZI_{snow}$ less than –1.0), and contiguous
areas under drought are shown and analyzed in Fig. 8, in which the vertical pink dashed lines mark the top five most
extensive droughts in each continent. We also selected three of the top five most extensive droughts to show their spatial
distribution (Fig. 9). The global averaged $SZI_{snow}$ displays a significant downward trend of –0.02 decade$^{-1}$ (95% confidence
level, Fig. 8a), which indicates a global drying trend. This drying trend was closely related to increases in temperature over
the study period. Accordingly, the global area under drought shows an upward trend (0.31% decade$^{-1}$) and approaches a
plateau over the period 1985–1995. It is clear that the contiguous area under drought demonstrates a similar pattern of
variability to the area under drought for each continent and globally. Such similarity implies the large-scale drought
identified by the SAD method can largely reflect the variability of the global area under drought.

Asia experienced a drying trend, based on the area-averaged $SZI_{snow}$, during the period 1948–2010 (Fig. 8b); the contiguous
area under drought ranges from 0% to 29.30%, with an average of 10.14%. With large fluctuation, droughts in early 1990s
are salient features of the time series of Asia, and three of the five droughts with the largest spatial extent occurred during the
1990s. The drying trend in east Asia was mainly caused by weakening summer monsoons owing to changes in the El Niño–
Southern Oscillation (ENSO) and the Pacific Decadal Oscillation (Zhang and Zhou, 2015). The large-scale severe droughts
in the Middle East and southwest Asia were closely related to La Niña (Barlow et al., 2016). Additionally, the temporal
variability within Asia is comparably small, mainly due to the dampening effect of its large spatial scale (Sheffield et al.,
2009). In Europe, high variability of the contiguous area in drought was detected in the first half of the 1950s (Fig. 8c). The
drought condition alleviated somewhat between the mid-1950s and the mid-1970s. The high variation was repeated in the
1990s, and was associated with multiple periods of droughts with large spatial extent. In particular, large-scale droughts
identified by the $SZI_{snow}$ occurred with a greater frequency over central Europe compared to other parts of Europe. The
leading driver behind this pattern was the significant increase in potential evapotranspiration (Spinoni et al., 2015a). The
findings in Europe, based on $SZI_{snow}$, are broadly in agreement with other studies (Lloyd-Hughes and Saunders, 2002;
Spinoni et al., 2015b).

In Africa, the area-averaged $SZI_{snow}$ exhibits a visible drying trend from 1948 to 2010 (Fig. 8d). The time series of drought
areas underwent a gradual climb and achieved a maximum value in the mid-1980s, with a severe drought period then lasting
until the mid-1990s. All the top five spatially extensive droughts are concentrated within this period and are commonly
located to the south of the Sahara Desert (Figs. 9g–9i). Our results for Africa are generally similar to previous studies, which
concluded that ENSO and sea surface temperature (SST) are the main driving forces of droughts across the entire continent
(Masih et al., 2014). For Oceania, strong drought spells occurred frequently from the 1950s to the 1970s (Fig. 8e). This
continent is characterized by its high percentage of large-scale drought areas, and multiple droughts account for more than
40% of the total continent (Figs. 9j–9l). The characteristics of historical droughts in Oceania are associated with variability



of global climate, for instance, the Interdecadal Pacific Oscillation (IPO) and Southern Annular Mode (Askarimarnani et al., 2021; Kiem et al., 2016).

In North America (Fig. 8f), the evident drought spells in the 1950s were captured by the $SZI_{snow}$, and the largest drought area covered 37% of the entire continent. As shown in Fig. 9o, the drought in March 1964 covered most of the United States. Previous studies confirmed that the tropical part of the SST anomalies was primarily related to the most notable droughts of

the 1950s in the United States (Schubert et al., 2004). Another two obvious drought signals are found in the late 1970s and 1990s. The droughts detected here with the $SZI_{snow}$ show close correspondence to the findings of previous studies (Su et al., 2021; Andreadis et al., 2005). Moreover, notable distinct dry spells emerged in the 1960s and 1990s in South America (Fig. 8g). For instance, the largest drought in October 1963 covered up to 54% of this continental area (Fig. 9r) and covered nearly the whole of Brazil. After a strong dry spell in 1998, South America exhibited a low percentage of drought extent until the

end of the studied time series.

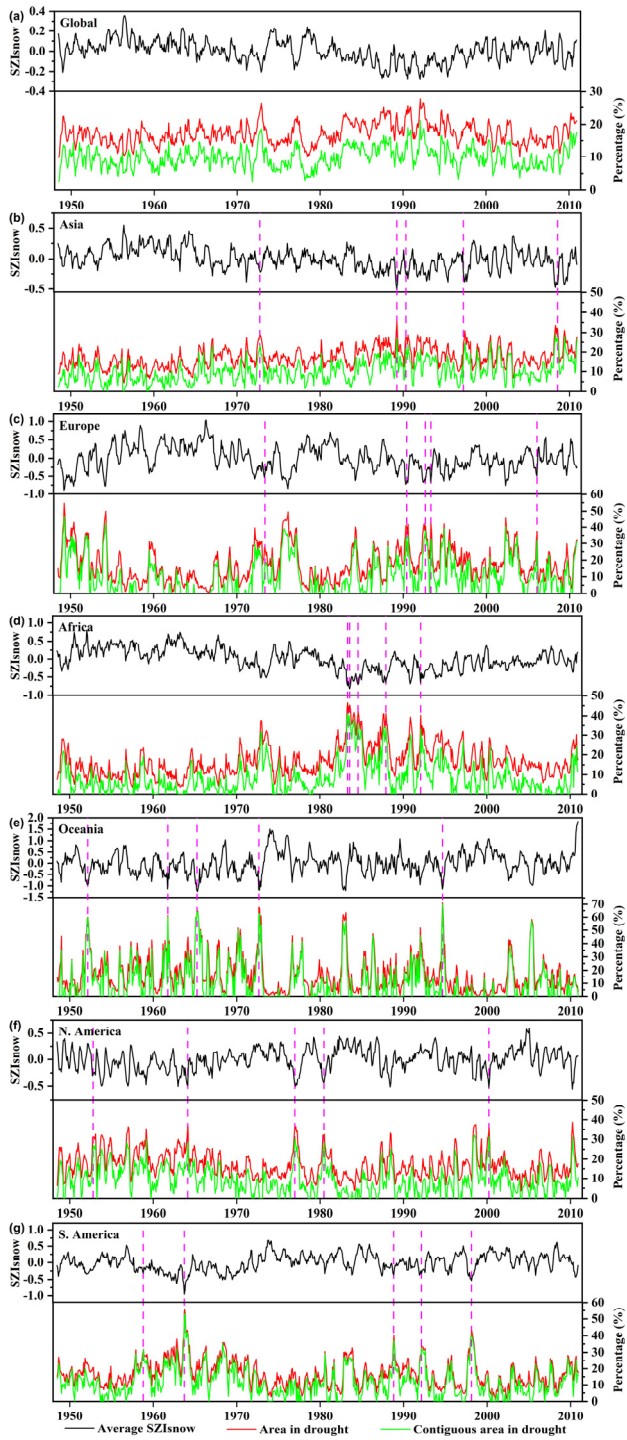

**Figure 8: Temporal variation of monthly area-averaged SZI$_{snow}$ (black lines), the area in drought (pixels with SZI$_{snow}$ < −1, red lines), and contiguous area in drought (green lines) for the world, Asia, Europe, Africa, Oceania, North America, and South America. The vertical pink dashed lines mark the top five major large-scale drought events in each continent.**




**Figure 9: Spatial distribution and severity of the major large-scale drought events for Asia, Europe, Africa, Oceania, North America, and South America. Three out of the top five drought events were selected here for each continent.**





## 5 Discussion and conclusions

This study proposes a drought index dataset on the basis of a new drought index, $SZI_{snow}$, by incorporating snow dynamics
into the SZI. Results from the evaluation of the $SZI_{snow}$ dataset suggest that consideration of snow processes can improve the
performance of the $SZI_{snow}$. The improvement is remarkable when the $SZI_{snow}$ is applied in snow-covered areas, including
high-latitude and high-altitude areas. Our results highlight the importance of snow in drought development because it can
greatly affect the onset, cessation, severity, location, and duration of drought (Huning and Aghakouchak, 2020; Staudinger et
al., 2014). Snow serves as the main water resource for many regions of the world (e.g., western United States) through its
accumulation in the cold season and melting in the warm season. However, climate change is altering the effect of snow on
the availability of water resources. Increasing temperature leads to less snowfall and earlier snowmelt, and further results in a
mismatch between the peak of streamflow and that of water demand, which can increase the drought risk over these regions
(Adam et al., 2009; Özdoğan, 2011). The results of the present work underscore the importance of considering snow
processes in drought quantification under global climate change.

Using the proposed $SZI_{snow}$ dataset, this study emphatically analyzed the severity–area–duration of global and continental
large-scale drought. The $SZI_{snow}$ dataset achieved a satisfactory performance in monitoring the propagation of large-scale
contiguous droughts through space and time. Using the SAD drought diagnosis method, the $SZI_{snow}$ dataset appropriately
captures the numbers and variability of historical large-scale contiguous droughts for each continent. These captured drought
events are broadly aligned with findings from previous research (Zhang and Zhou, 2015; Mctainsh et al., 1989; Kiem et al.,
2016; Lloyd-Hughes and Saunders, 2002). Such performance implies the present dataset can be applied globally to
understand the mechanisms behind large-scale droughts. It also raises confidence in the ability of the $SZI_{snow}$ to predict
drought events, especially those with extensive spatiotemporal influence. Moreover, our results indicate that large-scale
contiguous droughts control, to a large extent, the character of the variation of global drought. Thus, the capacity to track the
evolution of large-scale droughts in space and time is a crucial aspect for the assessment of a drought index.

The $SZI_{snow}$ absorbs the advantages of both the PDSI and SPEI and can be used to monitor multitype droughts at various
temporal scales. Compared to the PDSI, it considers more hydrological components related to water supply and demand, and
quantifies their contribution to water demand by weight. Such consideration enhances the physical realism of drought
quantification, particularly over high-latitude and high-altitude regions that usually receive substantial snowfall (Zhang et al.,
2019). The enhancement achieved by the $SZI_{snow}$ implies that more key physical processes should be considered when
constructing a drought index, rather than using a simple generalization, although we admit that a sophisticated index is
always limited by insufficient observation to some extent. However, data assimilation serves as a new way to overcome the
difficulty of insufficient observation (Mishra and Singh, 2011). This new method combines a multi-source dataset and an
advanced land surface model to provide optimal values of variables related to drought, which is the reason why we used
GLDAS-2 as the forcing means of $SZI_{snow}$ calculation. Therefore, the improvement of $SZI_{snow}$ indicates that more attention
should be paid to the combination between the drought index and the data assimilation system (DAS) or LSM.



The combination between the SZI$_{snow}$ and the DAS provides the possibility to track droughts over ungauged areas. As more models (e.g., crop model, wildfire model, root model) have been coupled with the DAS, the combination between the SZI$_{snow}$ and the DAS has become more physically realistic. Yet, uncertainties from the DAS will inevitably be introduced into the SZI$_{snow}$, which undermines the reliability of the SZI$_{snow}$. Previous studies have often obtained dissatisfactory results during

the validation of the GLDAS-2 (e.g., Fatolazadeh et al., 2020). These uncertainties originate from incomplete model structure, forcing data biases, and biases in parameter estimation (Qi et al., 2020). However, recent developments in LSMs, DAS techniques, and computational power are helpful in resolving issues associated with uncertainty. Thus, determining how to introduce uncertainty quantification when utilizing the SZI$_{snow}$ to assess drought is a future goal of ours.

The SZI$_{snow}$ is a comprehensive drought index because it incorporates different aspects of the hydrologic cycle, which

provides a clear-cut way to synthesize different kinds of information related to drought into a simple message. Such synthesizing capacity is particularly crucial because droughts have a broadly adverse influence on agricultural water, municipal water, energy supply (hydropower), and human and animal safety. Thus, the SZI$_{snow}$ has a high potential to be utilized for drought management. Currently, however, the SZI$_{snow}$ is mostly used only by the scientific community (Lu et al., 2020; Ayantobo and Wei, 2019) and rarely used by decision- and policy-makers. One reason for this is that the acquisition of

best-fit thresholds in the SZI$_{snow}$, for one type of drought over an area with a specific climate regime, requires a trial-and-error approach and takes time. On the other hand, drought management is a synergistic effort involving a variety of sectors and requires joint operations of these sectors. Additionally, the complexity of calculations is a limitation of the SZI$_{snow}$. Therefore, it is necessary to strengthen the user-friendliness of the SZI$_{snow}$ and collaborate closely with government departments related to drought management.

**6 Data availability**

All datasets used in this work are freely available. The SZI$_{snow}$ dataset proposed by this work is a good contribution to the study of climate change, ecology, and hydrology. It is especially helpful for research focusing on spatiotemporal dynamics of drought, the underlying mechanisms of drought evolution, and the development of drought indices. The dataset contains 48 individual files with timescales of 1–48 months and has been archived in the Network Common Data Form (NetCDF) format.

The monthly SZI$_{snow}$ in each file covers the Earth's land area and has a spatial resolution of 0.25°. The SZI$_{snow}$ dataset is freely downloadable from the Zenodo repository at the following URL: http://doi.org/10.5281/zenodo.5627369 (Wu et al., 2021). In addition, we also published the dataset to the National Tibetan Plateau/Third Pole Environment Data Center, which has been accredited by the Earth System Science Data, and specializes in collecting, integrating, and publishing geoscientific data on and surrounding the Tibetan Plateau and the three poles. The SZI$_{snow}$ dataset can be downloaded from this data center

at the following URL: http://data.tpdc.ac.cn/en/disallow/b039fde6-face-4d24-af45-d238a6af18b7/.

## 7 Summary

In the current study, we have produced a global monthly $SZI_{snow}$ dataset over 1–48 month timescales from 1948–2010. This dataset is an important contribution to drought quantification and development of drought indices because it is built on the $SZI_{snow}$, a multitype and multiscalar drought index absorbing the strengths of the SPEI and PDSI. Our $SZI_{snow}$ dataset has achieved a remarkable improvement in drought assessment across the world, particularly for high-latitude and high-altitude areas. This improvement implies that consideration of snow processes can improve the performance of a drought index. Moreover, the $SZI_{snow}$ dataset can successfully monitor the spatiotemporal propagation of large-scale drought events. We expect this dataset could serve as a valuable resource for drought studies, further contributing to promoting our understanding of the mechanisms behind global drought dynamics.

**Author contributions**

BZ and PW designed the research and developed the methodology; LT, BZ, and PW supervised the processing of the datasets; LT conducted the statistical analysis and wrote the manuscript; LT, BZ, and PW revised the manuscript.

**Competing interests**

The authors declare that they have no conflict of interest.

**Acknowledgements**

We acknowledge the observed hydrologic data developed for 32 global basins provided by Pan et al. (2012), and the SWI data provided by Liu et al. (2017). We would also like to thank the National Aeronautics and Space Administration (NASA) for their support in providing the GLDAS data. Thanks to Dr. Akash Koppa and Dr. Dominik Rains for producing and providing the GLEAM data. Finally, we thank the Climatic Research Unit (University of East Anglia) and the UK National Centre for Atmospheric Science (NCAS) for providing the CRU TS dataset.

**Financial support**

This research was supported by the National Natural Science Foundation of China (Grant Nos. 42041004, 42022001, 42001029 & 41877150) and the National Key R&D Program of China (Grant No. 2020YFA0608403).




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
