# Peer review of "A global drought dataset of standardized moisture anomaly index incorporating snow dynamics (SZIsnow) from 1948 to 2010"

_Earth System Science Data, 2021_

## Referee Comment (RC4)

**A global dataset of standardized moisture anomaly index incorporating snow dynamics from 1948 to 2010**

**Overall comments**

This manuscript developed a new global monthly drought index dataset with multi-types and multi-scales, SZIsnow. The drought index SZIsnow incorporates different physical water–energy processes with snow process.The dataset also was comprehensively evaluated by different drought types, different spatial scales. The drought index SZIsnow and SZI are compared in different regions.

The dataset can serve as a valuable resource for drought studies. The paper is scientifically sounding. The topic well fits the scope of this special issue. The manuscript is well written and logically organized and the dataset was easy to access. However, some concerns still need to be addressed and make it clearer for the readers before publication. Below are my several comments.

**General Comments**

1. The title used "standardized moisture anomaly index",however,"drought index" is used more in the text. Need to consider a better title to attract the interest of the potential data users.

2. The abstract does not show the spatial resolution of your dataset. It is an essential parameter for reader and data user.

3. In the Line 20, "Our results also show that the SZIsnow dataset successfully captured the largescale drought events that occurred across the world; there were 525 drought events with an area larger than 500,000 km2 globally during the study period, of which nearly 70% had a duration longer than 6 months." What is the accuracy rate of this product? How to evaluate this more reasonable? The product capture all the drought events? Is its capture rate 100%?

4. In the Figure 1. I did not see the description of scPDSI in the manuscript.

5. I suggest adding the description of the advantages of the SZIsnow in the figure 1.

6. The GLDAS-2 data provide the variables to calculate the SZIsnow from 1948-2010. GLDAS-2.1 is one of two components of the GLDAS Version 2 (GLDAS-2) dataset, from 2000 to present. Is it possible to use GLDAS-2.1 to extend the time coverage of the product?

7. Section 2.4 "Metrics for the SZIsnow evaluation" is not the data. It is an accuracy assessment method, not the data description. Is it more appropriate to move this part to the section 3.

8. Line 164 "The prominent improvement of the SZIsnow is that it accounts for the influence of snowfall on hydrological processes, which was completely ignored in the SZI (Zhang et al., 2019; Zhang et al., 2015)."

   Line 171 "Both the soil moisture storage and snow storage are considered as reservoirs in the SZIsnow, which is different from the SZI that solely considered the former."

   This section is to discuss how to produce the SZIsnow. The difference between SZI and SZIsnow should be placed in the validation section.

9. I suggest a procedure flowchart describing the production and validation of SZIsnow. It would be better that the advantages of the SZIsnow are mentioned in the figure. The flowchart can facilitate users to understand the dataset.

---

## Author Comment (AC1)

Response to Referee #1:

The authors would like to thank you for the constructive and thoughtful comments. We have addressed all your suggestions, leading to a much improved and complete manuscript. The following comments are addressed in the sequence as they were asked. We also respond to each point and clarify the corresponding changes adopted in the revised manuscript. The original comments are copied from the report with a **bolded font in black**, and our answers are in blue. Manuscript changes are in ***bold italic***.

Thank you again for your time and effort in reviewing our manuscript.

Yours sincerely

Baoqing Zhang (on behalf of all co-authors)

**Reviewer #1**

This manuscript introduces a drought product with a multitype and multiscalar drought index, $SZI_{snow}$. The $SZI_{snow}$ dataset considers a relatively comprehensive extent of the hydrometeorological variables associated with drought development. It uniquely incorporates snow processes in the derivation of a drought index. This consideration is important as global warming has been affecting hydrological processes over the snow-covered regions. The dataset was evaluated across different spatial scales. The result shows the $SZI_{snow}$ has good performance over the snow-covered regions. The dataset was also used to survey the evolution of large drought events with the severity-area-duration method.

The topic of the study is interesting and well fits the scope of the journal, especially for this special issue. The manuscript is well written, logically organized, and the details of the derivation of the $SZI_{snow}$ are easy to follow. The data processing is careful and well documented, and the dataset was friendly to access. However, there are still some concerns that need to be addressed. Thus, I am supportive of the publication after a minor revision to further improve the quality or make it more clear for the readers to understand the results. Below are my suggestions:

Reply: We appreciate the reviewer's positive comments and interest in our work.

**General comments**

**#1 The main improvement in the $SZI_{snow}$ is the consideration of snow related processes, thus it is reasonable that the $SZI_{snow}$ in the snow-covered basins have a better performance than that of SZI as shown in Figure 3. However, I found that both the $SZI_{snow}$ and SZI have a similar performance over the snow-free basins. Please clarify this similar performance.**

Reply: We agree with this comment that the $SZI_{snow}$ has a similar performance with the SZI over the snow-free basins. Such similar performance is mainly attributed to the fact

that Psnow, PSM, and PSA values are close to zero over basins at low-latitude and low-altitude snow-free areas, and thus the calculation of SZI$_{snow}$ converges to the snow-free basins. Therefore, the performance of SZI$_{snow}$ and SZI are consistent with each other over snow-free areas.

For the SZI$_{snow}$:

$$\begin{cases} P = P_{rainfall} + P_{snowfall} \\ Z_{snow} = P - \hat{P}_{snow} \\ \hat{P}_{snow} = \alpha_j PET + \beta_j PR + \gamma_j PRO + \delta_j PSA - \varepsilon_j PL - \varphi_j PSM \end{cases}$$

For the SZI:

$$\begin{cases} Z = P_{rainfall} - \hat{P} \\ \hat{P} = \alpha_j PET + \beta_j PR + \gamma_j PRO - \delta_j PL \end{cases}$$

Following the comments, we will add some information into the discussion section of the revised manuscript to explain why the performance of the SZI$_{snow}$ is similar to that of the SZI in snow-free areas as follows:

*Besides the overperformance of the SZI$_{snow}$, it should be noted that the SZI$_{snow}$ has a similar performance with SZI over the snow-free basins. Such similar performance is mainly owing to the fact that the values of P$_{snow}$, PSM, and PSA are close to zero over basins at low-latitude and low-altitude (snow-free) areas, leading to the calculation of SZI$_{snow}$ converges to the snow-free basins. Therefore, the performance of SZI$_{snow}$ and SZI are consistent with each other over snow-free areas.*

**#2 The evaluation of the SZI$_{snow}$ is important compared to current drought indices (for example the scPDSI). Besides the SPI, I think the scPDSI is also a good index to assess the meteorological drought. Thus, it is necessary to evaluate the performance of SZI$_{snow}$ to capture meteorological drought compared with other indices instead of only with SPI. This can help to confirm the robustness of your conclusion.**

Reply: Following the comments, in addition to the SPI, we adopted two other mainstream drought indices (SPEI and scPDSI) to compare their performance in monitoring meteorological drought. Figure R1 shows the responses of multiple drought

indices to meteorological drought. As we can see, the performance of SZI$_{snow}$ is prominent and superior to SZI, SPEI, and scPDSI in identifying meteorological drought at multiple temporal scales. This further confirms our prior conclusion.

Following the comments, we will add the following information into section 4.1.1 of the revised manuscript, and Figure R1 will be added as Figure S1 into the supplementary material:

*In addition to the SPI, we adopted two other mainstream drought indices (SPEI and scPDSI) to compare their performance in monitoring meteorological drought. As shown in Figure S1, the performance of SZI$_{snow}$ is prominent and superior to SZI, SPEI, and scPDSI in identifying meteorological drought at multiple temporal scales. The selection of reference drought indices did not influence the reliability of our conclusion.*

[Figure]

**Figure R1.** The Pearson correlation coefficients between SPEI and SPI (i.e., SPEI-SPI), scPDSI-SPI, SZI-SPI, and SZI$_{snow}$-SPI at 1- to 48-month time scales in the selected 32 large basins during 1948–2010. The dominant climate regime of each basin is shown in the parentheses.

**Specific comments**

**#1 Is it possible to include the acronym of your new drought index in the title? Such inclusion can enhance the recognizability of your dataset and facilitate others to cite and employ your dataset.**

Reply: The title will be changed to "*A global dataset of standardized moisture anomaly index incorporating snow dynamics (SZI$_{snow}$) from 1948 to 2010*" in the revised manuscript.

**#2 I suggest adding the spatial resolution of your dataset in the Abstract section.**

Reply: The spatial resolution of a dataset is essential for the Earth System Science Data journal. The spatial resolution of our proposed dataset is 0.25 degrees. This content will be added in the revised manuscript as follows:

*Here, we present a global monthly drought dataset with a spatial resolution of 0.25° from 1948 to 2010 based on a multitype and multiscalar drought index, the standardized moisture anomaly index incorporating snow dynamics (SZI$_{snow}$), driven by systematic fields from an advanced data assimilation system.*

**#3 The author listed various current drought indices in Figure 1. Please make sure the corresponding references of these indices are supplied in the manuscript.**

Reply: Some drought indices in the first row of Figure 1 do not have corresponding references because those indices (e.g., SAI and PHDI) are not mentioned in the main text of the manuscript. Thus, we will add references of these non-mentioned indices in the supplementary material.

**#4 Line 38: Correct the "focus" to "focuses".**

Reply: This grammar problem will be corrected in the revised manuscript. Such problems have been fixed throughout the revised manuscript.

**#5 Line 288: Did the two subplots in Figure S3 have identical contour levels and color bar? If did, please remove one color bar.**

Reply: Yes, they did. We have removed one color bar of Figure S3, and the new Figure S3 will be added in the revised supplementary material as follows:

[Figure]

*Figure S3. Comparison between the SZI and SZI_snow in the context of their performance over the Arctic region. (a) Spatial distribution of the correlation coefficients of the SZI_snow–SWI over a 12-month timescale. (b) Spatial distribution of the correlation coefficients of the SZI–SWI over a 12-month timescale.*

**#6 Line 144: Consider adding a comma after the introductory phrase "meanwhile".**

Reply: This grammar problem will be corrected in the revised manuscript.

**#7 Line 300: What does the shading in the Figure 4d panel mean? It should be clarified in the Figure caption. The inset of the Figure 4d panel seems not clear to me, you can increase the resolution of your figure.**

Reply: The shading in Figure 4d denotes the range of correlation coefficients of the SZI_snow–SWI and SZI–SWI. The upper (lower) boundary is the maximum (minimum)

value. In addition, the original resolution of Figure 4 is 300 dpi, and the new Figure 4 has a resolution of 600 dpi, which makes it clearer. An explanation of the shading will be added into the caption of Figure 4 in the manuscript as follows:

[Figure]

*Figure 4: Performance of the SZIsnow over different latitudes (a and b) and specifically over the Arctic region (c and d). Here the differences between the correlation coefficients of the SZIsnow–SWI and those of the SZI–SWI for different timescales were used to 300 compare their performance. (a) The Hovmöller diagram (timescale ⊙⊙⊙ latitude) shows the differences averaged by latitude from 55˚S to 85˚N for timescales ranging from 1 to 48 months. (b) Distribution of the difference for specific timescales (6, 9, 12, and 15 months) with changing latitude. (c) Spatial distribution of the differences between the correlation coefficients of the SZIsnow– SWI and those of the SZI–SWI over a 12-month timescale in the Arctic region. (d) Variations of correlation coefficients averaged over the Arctic region for various*

*temporal scales. The shading denotes the range of correlation coefficients. The upper (lower) boundary is the maximum (minimum) value. The inset shows the change of relative difference (%) for these temporal scales.*

**#8 Line 315: Did the subplots in the left and middle columns of Figure S3 have identical contour levels and color bar? If did, please remove one color bar.**

Reply: Same as comment #5, this comment suggests that we need to remove the identical units in a figure to make it more concise; thus we examined similar problems across all the figures and found Figure 5 has such a problem. We corrected this problem of Figure 5 in the revised manuscript as follows:

[Figure]

*Figure 5: Spatial distribution of correlation coefficients of the $SZI_{snow}$–SWI (left column) and those of the SZI–SWI (middle column), and the differences between the two (right column = left column minus middle column) over the Tibetan Plateau at different timescales (6, 9, 12, and 15 months).*

**#9 Line 330: Some corner strings (e.g. "a", "b" in panels) are bolded, but some aren't. Please keep the format of these strings consistent across the manuscript.**

Reply: We checked and modified the appearance of corner strings throughout the manuscript. As a result, the format of these strings is now consistent across the revised manuscript.

**#10 I recommend adding the values of trends in Figure 7 so that the reader can know more information from the figure.**

[Figure]

*Figure 7: Time series of (a) global dry land area (% yr$^{-1}$) and (b) $Z_{snow}$ (mm yr$^{-1}$) between 1948 and 2010. The dry land area was calculated based on the SZI$_{snow}$ at a 12-month timescale. The dashed lines denote the linear trends, and the numbers represent the value of change rate.*

**#11 Line 352: It is a good way to use a country to describe the area of a drought event. Please add a number to show how large is the size of Guatemala.**

Reply: The area of Guatemala is 108,889 km² that will be added in the revised manuscript as follows:

*The most spatially extensive drought occurred over Asia in August 2008 (drought lasted from November 2007 to June 2009) and covered an area of approximately 11 million km² (roughly 100 times Guatemala's national territory area of 108,889 km²).*

**#12 Line 400: The Interdecadal Pacific Oscillation just appears one time, it is not necessary to provide an abbreviation for it. Please correct the same problems throughout the manuscript.**

Reply: The abbreviation of the Interdecadal Pacific Oscillation (i.e., IPO) will be removed in the revised manuscript.

**Dataset**

**I downloaded the compressed files including all the files from http://doi.org/10.5281/zenodo.5627369. With the software of "ncdump" and "Panoply", I checked the data files and had no problems to read and visualize the data. All the data is consistent as they were described in the manuscript. Here I give some recommendations to improve the user-friendliness of the proposed dataset.**
**#1 The compressed file with a suffix of ".zip". Is it possible add an introduction about how to unzip this kind of file?**

Reply: The files in the Zenodo repository were compressed by a free and open-source file archiver, 7-Zip, due to its higher compression ratio than other common software. Following the comment, we will add below introduction about how to unzip this kind of file in the Metadata file:

*All the files were compressed by the 7-Zip software. The 7-Zip is a free and open-source file archiver and can be downloaded from the URL: https://www.7-zip.org/download.html. After opening the URL, you can choose a suitable install file based on your operating system. The operation of 7-Zip is similar to the standard file archiver.*

**#2 The SZIsnow datasets have different timescales, thus, months before the timescale are set as missing values. It would be nice to give a clear introduction about the missing values in your dataset.**

Reply: Following the comment, we will add below introduction about the missing values of the $SZI_{snow}$ dataset in the Metadata file:

*The $SZI_{snow}$ is a multiscalar drought index, and the $SZI_{snow}$ variable in the data file has different timescales. Thus, for a $SZI_{snow}$ variable (len, time, lat, lon) at n months timescale, the first n-1 months (from 1 to n-1) in the SZIsnow variable are missing values.*

---

## Author Comment (AC2)

Response to Referee #2:

The authors would like to thank you for the constructive and thoughtful comments. We have addressed all your suggestions, leading to a much improved and complete manuscript. The following comments are addressed in the sequence as they were asked. We also respond to each point and clarify the corresponding changes adopted in the revised manuscript. The original comments are copied from the report with a **bolded font in black**, and our answers are in blue. Manuscript changes are in ***bold italic***.

Thank you again for your time and effort in reviewing our manuscript.

Yours sincerely

Baoqing Zhang (on behalf of all co-authors)

**Reviewer #2**

**The authors developed a global dataset of standardized drought indicators incorporating snow information. They first evaluated the indicator and then employed it for the drought temporal-spatial analysis across the globe. This is an interesting dataset for the drought analysis at the global scale. I have several comments as follows.**

Reply: We appreciate the reviewer's comments and interest in our work.

**General comments**

**Figure 1 In the box of hydrological drought. The advantage (green box) is "considers snowpack and water storage" and the disadvantage (red box) is "unsatisfactory performance over snow covered regions". What is the reason for the "unsatisfactory performance over snow…" if these indices already consider "snow pack"?**

Reply: We would like to apologize for this vague description. We list three drought indices in the box of hydrological drought (i.e., SDI, PHDI, and SWI). The streamflow drought index (SDI) only requires monthly streamflow for its calculation. The Palmer hydrological drought severity Index (PHDI) requires monthly temperature, precipitation, and water-holding capacity of soils for its calculation. Thus, the SDI and PHDI have unsatisfactory performance over snow-covered regions because the input parameters of the two indices (SDI and PHDI) do not include parameters related to snow.

In the three listed indices in the box of hydrological drought, only the SWI (surface water supply index) considers snowpack in its calculation. Its input parameters include streamflow, precipitation, reservoir storage, and snowpack. Nevertheless, the SWI is basin-dependent by calculating at the basin level, and it is difficult to compare basins, not to mention at the grid level.

To clarify this vague description, we will update Table S1 in the supplementary material, which will clearly list the strengths and weaknesses of each drought index shown in Figure 1.

**Figure 1 It seems the SZI addresses all the challenges of the indices mentioned in this figure. Is there any remaining disadvantage or limitation of SZI? Please clarify.**

Reply: We agree with this comment and admit that the SZI can not address all the challenges of the indices mentioned in this figure. The SZI does still have some remaining disadvantages or limitations. Figure 1 is used to show our motivation and train of thought for the development path of the index. We thus did not put too much emphasis on the limitations of the SZI in the original manuscript. In general, there are two main limitations of the SZI. The first one is that its computation is more complex and difficult than the SPI or SPEI. Another limitation is that its calculation requires long-term climatic and hydrologic records, making it unsuitable for short-term drought identification and monitoring.

Following the comments, we will add the following information into the fourth paragraph of Introduction Section in the revised manuscript:

*Additionally, there are two main limitations of the SZI. The first one is that its computation is more difficult than the SPI or SPEI. Another limitation is that it needs a long-term serial of hydrometeorological records, making it unsuitable for short-term drought studies.*

**Lines 133: "We evaluated the ability of the SZIsnow and SZI to capture different" It is generally hard for a single drought indicator to capture all types of droughts. Is the SZI designed to capture all drought types?**

Reply: We cannot agree more with this comment. It is generally hard for a single drought indicator to capture all types of droughts. Indeed, it is essential to realize this

challenging objective. This is because the impacts of droughts are significant and widespread, affecting many economic sectors and people at any one time. In addition, different types of droughts are considered to be interchangeable. Drought can convert from one type to another as it evolves in time and space. Nevertheless, different administrative departments nowadays employ various drought indices for drought management, leading to unaligned action plans against drought. Without alignment, there is likely to be considerable delay in action at the onset of drought in an area or region. Therefore, our study strives to design a drought index, SZI, to capture all drought types. Though it is challenging to develop a multitype index and our proposed index is not 100-percent perfect, our work is on the right track. In this way, different sectors of society can collaborate to synergistically fight against drought using a comprehensive drought index.

**Lines 201-202: Here P and Psnow are used to define the water supply deficit. This equation is only applied in regions and seasons with snowfall, right? What about other regions (e.g., tropics)? Do you use a different set of equations to calculate SZI?**

Reply: Our equation can be applied in regions and seasons with and without snowfall. For the regions without snowfall (e.g., tropics), the items relevant to snow in the equations are set to zero for the calculation of SZI$_{snow}$. For example, in the following equation, the $P_{snowfall}$, $\delta_j PSA$, and $\varphi_j PSM$ are set to zero when they are used for situations without snowfall.

$$\begin{cases} P = P_{rainfall} + P_{snowfall} \\ Z_{snow} = P - \hat{P}_{snow} \\ \hat{P}_{snow} = \alpha_j PET + \beta_j PR + \gamma_j PRO + \delta_j PSA - \varepsilon_j PL - \varphi_j PSM \end{cases}$$

This comment is an important reminder for our study. Following the comments, we will add some information into section 3.1 of the revised manuscript to explain how to calculate SZI$_{snow}$ for regions and seasons without snowfall:

*In addition, our equations can be applied in regions and seasons without snowfall. For regions without snowfall (e.g., tropics), the items relevant to snow in the equations of Table 1 are set to zero for the calculation of $SZI_{snow}$. For example, $\delta_j PSA$ (Equation 8), $\varphi_j PSM$ (Equation 8), and $P_{snowfall}$ (Equation 9) are set to zero when they are used for situations without snowfall.*

**Lines 244-245: Here the authors used SPI to evaluate the proposed index SZIsnow. SPI mainly reflects precipitation-related droughts. There may not be snow information in SPI (it does not incorporate snowfall, right?). How do we know a higher SZIsnow-SPI correlation reflects better performance of the proposed index? Please justify this.**

Reply: The standardized precipitation index (SPI) uses historical precipitation records for any location to develop a probability of precipitation that can be computed at any number of timescales, from 1 month to 48 months or longer. The only input parameter of SPI is precipitation, which is the greatest strength of SPI. Precipitation is regarded as the sum of rainfall and snowfall during the calculation of SPI. Thus, the SPI considers the snow information, and it is reasonable to use SPI for the evaluation of the $SZI_{snow}$.

**Lines 294-295: It seems the proposed index performs better for long-time scales. Any specific reason for this? Please clarify.**

Reply: We agree with this comment that the proposed index performs better for long-time scales. There are mainly three reasons for its better performances at long timescales. Firstly, the magnitude of the accuracy of the moisture anomaly $Z_{snow}$ (Equation 9 of Table 1) is small at a short timescale (e.g., 1-month). In comparison, the accuracy of $Z_{snow}$ becomes relatively larger at a long timescale (e.g., 12-month) because the 12-month $Z_{snow}$ on a certain date is an accumulative value by summing up monthly $Z_{snow}$ of the prior 12 months. Accordingly, the improvement is more evident at a long timescale. Secondly, it usually takes a long time from the arrival of precipitation (rainfall and snow) to precipitation becomes different forms of useable water.

Precipitation infiltrates into the groundwater takes longer than it directly converts to overland flow. In addition, the snowpack accumulates in the cold season, and snowmelt drains into the soil or directly into the river channel in the warm season, which leads to a several-month to 1-year lag response in the soil wetness and total water storage variability. At last, long-term droughts always cause more widespread and severe consequences than short-term droughts. The capacity of the $SZI_{snow}$ to identify and monitor long-term drought is the focus of our study.

**Figure 6. This trend analysis for each grid is performed for the annual time series? Or for a season? Please clarify.**

Reply: We performed the trend analysis for each grid in Figure 6 for the annual time series. Following the comments, we will provide a clearer caption and description for Figure 6 in the revised manuscript as follows:

*Figure 6: Spatial distribution of the linear annual trend (changes per 50 years) in the $SZI_{snow}$ during the period 1948–2010, at various timescales. The stippling denotes the trend being statistically significant at the 95% confidence level.*

*The spatial distribution of the linear annual trend in the $SZI_{snow}$ over different timescales (i.e., 3-, 6-, 12-, 15-months) is shown in Fig. 6.*

**Figure 7 Any speculation for the low SZI and high dry areas during 1985-1990?**

Reply: The $SZI_{snow}$ employs precipitation amount (referred as $\widehat{P}_{snow}$) that is climatically appropriate for existing conditions to quantify the regional water demand. The moisture anomaly $Z_{snow}$ is defined as the difference between the actual Precipitation (P) and $\widehat{P}_{snow}$, which is an appropriate indicator for regional water deficiency or surplus. In Figure 7b, we used $Z_{snow}$, instead of $SZI_{snow}$, to represent dry and wet conditions over a region. There is a negative correlation between $Z_{snow}$ and dry areas. Thus, the high dry areas correspond with the low $Z_{snow}$ during 1985-1990.

---

## Author Comment (AC3)

Response to Referee #4:

The authors would like to thank you for the constructive and thoughtful comments. We have addressed all your suggestions, leading to a much improved and complete manuscript. The following comments are addressed in the sequence as they were asked. We also respond to each point and clarify the corresponding changes adopted in the revised manuscript. The original comments are copied from the report with a **bolded font in black**, and our answers are in blue. Manuscript changes are in ***bold italic***.

Thank you again for your time and effort in reviewing our manuscript.

Yours sincerely

Baoqing Zhang (on behalf of all co-authors)

**Reviewer #4**

**Overall comments**

**This manuscript developed a new global monthly drought index dataset with multi-types and multi-scales, SZIsnow. The drought index SZIsnow incorporates different physical water–energy processes with snow process. The dataset also was comprehensively evaluated by different drought types, different spatial scales. The drought index SZIsnow and SZI are compared in different regions.**

**The dataset can serve as a valuable resource for drought studies. The paper is scientifically sounding. The topic well fits the scope of this special issue. The manuscript is well written and logically organized and the dataset was easy to access. However, some concerns still need to be addressed and make it clearer for the readers before publication. Below are my several comments.**

Reply: We appreciate the reviewer's comments on our work.

**General Comments**

**1. The title used "standardized moisture anomaly index", however, "drought index" is used more in the text. Need to consider a better title to attract the interest of the potential data users.**

Reply: We thank the reviewer for this excellent comment. The title will be changed to "*A global drought dataset of standardized moisture anomaly index incorporating snow dynamics (SZI$_{snow}$) from 1948 to 2010*" in the revised manuscript.

**2. The abstract does not show the spatial resolution of your dataset. It is an essential parameter for reader and data user.**

Reply: We cannot agree more with this comment. The spatial resolution of a dataset is an essential parameter for readers and data users. The spatial resolution of our proposed dataset is 0.25 degrees. This content will be added in the revised manuscript as follows:

*Here, we present a global monthly drought dataset with a spatial resolution of 0.25°*
*from 1948 to 2010 based on a multitype and multiscalar drought index, the*
*standardized moisture anomaly index incorporating snow dynamics (SZI$_{snow}$), driven*
*by systematic fields from an advanced data assimilation system.*

**3. In the Line 20, "Our results also show that the SZIsnow dataset successfully captured the largescale drought events that occurred across the world; there were 525 drought events with an area larger than 500,000 km$^2$ globally during the study period, of which nearly 70% had a duration longer than 6 months." What is the accuracy rate of this product? How to evaluate this more reasonable? The product capture all the drought events?Is its capture rate 100%?**

Reply: We thank the reviewer for this valuable comment. We recognize that the proposed product can not capture all the largescale drought events, and the capture rate is not 100%. The accuracy of drought assessment is a challenge for drought study, primarily because there did not exit an indicator to quantify drought directly. The lack of long-term observation is also the main reason for the difficulty of drought assessment. In addition, incomplete model structure, forcing data biases, and biases in parameter estimation in the forcing dataset (GLDAS-2) of the SZI$_{snow}$ can lead to the inaccuracy of our results. Furthermore, the clustering algorithm of the SAD method allows for the merging of two or more sub-droughts into a drought. Although the smooth and continuous movement of the drought clusters hints at some common underlying mechanisms, it is sometimes difficult to interpret sub-droughts as a single event due to likely different forcing mechanisms behind them. From the above considerations, it is hard to give an accurate rate of the captured droughts.

To the best of the authors' knowledge, the reasonable approach for evaluating our results is the comparison with the published paper in the science community. As shown in Section 4.3.2, we compared our captured events with those documented by other studies for each continent. Although these documented drought events were identified based on different drought indices, these comparisons indicate that our captured

drought events are broadly aligned with findings from previous research. For example, Sheffield et al. (2009) pointed out that there have been 296 droughts greater than 500,000 km$^2$ globally from 1950 to 2000, with a dataset of soil moisture from simulations using the variable infiltration capacity model. This number is 311 in our result during the same period.

We recognize that the sentence in Line 20 is not an appropriate description of our results. Therefore, following the comment, this sentence will be corrected in the revised manuscript as follow:

*Our results also indicate that the SZI$_{snow}$ dataset can be employed to capture the largescale drought events that occurred across the world. Our analysis shows there were 525 drought events with an area larger than 500,000 km$^2$ globally during the study period, of which nearly 70% had a duration longer than 6 months.*

**4. In the Figure 1. I did not see the description of scPDSI in the manuscript.**

Reply: As it is known, the self-calibrated PDSI (scPDSI) is developed based on the original PDSI and accounts for all the constants contained in PDSI. The scPDSI includes a methodology in which the constants are calculated dynamically based upon the characteristics present at each station location. Thus, the scPDSI can generate more representative model constants at temporal and spatial scales. In Figure 1, as the methodology is not significantly different from PDSI, it has the same issues as the PDSI in terms of the fixed temporal scale and poor performance over snow-covered areas. Therefore, we did not mention scPDSI in the manuscript.

Following the comment, we will add the below content in the revised manuscript:

*In addition, the self-calibrated PDSI (scPDSI) can compute dynamically the constants in PSDI on the basis of the characteristics at each interested location, producing more representative model constants. However, the scPDSI has the same issues as the PDSI in terms of the temporal scale and performance over snow-covered areas.*

**5. I suggest adding the description of the advantages of the SZIsnow in the figure 1.**

Reply: Following the comment, we will add the description of the advantages of the SZI$_{snow}$ in Figure 1 of the revised manuscript:

[Figure]

**Figure 1.** Development path of the SZI$_{snow}$. Dark green boxes denote the strengths of each drought index, while pink boxes denote the weaknesses of each drought index. The top row shows indices that can only account for one type of drought, with three indices listed for each type of drought. The second row shows indices that can account for multiple types of drought. Full names of the listed indices are shown in Table S1.

**6. The GLDAS-2 data provide the variables to calculate the SZIsnow from 1948-2010. GLDAS-2.1 is one of two components of the GLDAS Version 2 (GLDAS-2) dataset, from 2000 to present. Is it possible to use GLDAS-2.1 to extend the time coverage of the product?**

Reply: We thank the reviewer for this valuable suggestion. The time coverage of a drought index product is essential for drought study. The GLDAS-2.0 and GLDAS-2.1 are two components of the GLDAS Version 2 dataset (GLDAS-2). Although the GLDAS-2.0 is analogous to GLDAS-2.1, there are some differences in their meteorological forcing datasets. Thus, the performance of these two components might be different for a specific region. For example, a previous study found that different performances of GLDAS2.0 and GLDAS2.1 exists in runoff simulations over the Tibetan Plateau, which may be due to their different uncertainty in the forcing data (Qi et al., 2018). Moreover, the moisture anomaly ($Z_{snow}$) in $SZI_{snow}$ can be aggregated at different temporal scales, causing the former months have influences on the latter months. An abrupt jump can be found in the time series of $SZI_{snow}$ when we connect these two different forcing datasets, which will inevitably introduce systematic bias in the $SZI_{snow}$. Based on the above considerations, we temporarily did not extend the time coverage of the $SZI_{snow}$ dataset using the GLDAS-2.1 product. In further work, this suggestion will be taken full account. We will appraise these two components and develop an approach to balance their differences, extending the time coverage of the $SZI_{snow}$ dataset.

Qi, W., Liu, J., & Chen, D. (2018), Evaluations and Improvements of GLDAS2.0 and GLDAS2.1 Forcing Data's Applicability for Basin Scale Hydrological Simulations in the Tibetan Plateau, *Journal of Geophysical Research: Atmospheres*, 123(23), 13,128-113,148, https://doi.org/10.1029/2018JD029116.

**7. Section 2.4 "Metrics for the SZIsnow evaluation" is not the data. It is an accuracy assessment method, not the data description. Is it more appropriate to move this part to the section 3.**

Reply: Following the comment, we will move this section to Section 3 as Section 3.2 in the revised manuscript.

**8. Line 164 "The prominent improvement of the SZIsnow is that it accounts for the influence of snowfall on hydrological processes, which was completely ignored in the SZI (Zhang et al., 2019; Zhang et al., 2015)."**
**Line 171 "Both the soil moisture storage and snow storage are considered as reservoirs in the SZIsnow, which is different from the SZI that solely considered the former."**
**This section is to discuss how to produce the SZIsnow. The difference between SZI and SZIsnow should be placed in the validation section**

Reply: We should focus on how to produce the $SZI_{snow}$, instead of comparing the SZI and $SZI_{snow}$, since Section 3.1 discusses the derivation of the $SZI_{snow}$. We checked this section thoroughly to avoid a similar problem following the comment. All the similar sentences will be moved into the evaluation section to explain the different performances between the SZI and $SZI_{snow}$.

**9. I suggest a procedure flowchart describing the production and validation of SZIsnow. It would be better that the advantages of the SZIsnow are mentioned in the figure. The flowchart can facilitate users to understand the dataset.**

Reply: We cannot agree more with this comment. A procedure flowchart can facilitate users to understand the production and validation of the $SZI_{snow}$ dataset.
Following the comment, we will add the below figure and its corresponding text in Section 3 of the revised manuscript:

*We provide a procedure flowchart as shown in Figure 3 to show the production and validation of the $SZI_{snow}$. There are four steps for $SZI_{snow}$ production: hydrologic accounting, climatic coefficients, water demand, and standardization. Hydrologic accounting is to calculate the monthly six components relevant to the local water budget; Climatic coefficients are the weighting factors of these components for the calculation of the local water demand; The local water demand in the $SZI_{snow}$ is represented by the precipitation that is climatically appropriate for existing conditions (CAFEC, referred as $\widehat{P}_{snow}$); The last step is the standardization of the moisture anomaly ($Z_{snow}$), which is the difference between the actual precipitation (rainfall and snowfall). After achieving the global $SZI_{snow}$ dataset, we not only validated $SZI_{snow}$ for identifying different types of drought at basin scale, but also across different regions worldwide, especially snow-covered regions, at grid scale.*

[Figure]

**Figure 3.** The procedure flowchart describing the production and validation of SZI$_{snow}$. Variables derive the SZI$_{snow}$ from the GLDAS-2 (or other LSM and DAS). The production of SZI$_{snow}$ includes four steps. The SZI$_{snow}$ is validated at basin scale for three types of drought and at grid scale across different regions worldwide, respectively. The cloud-shape annotation shows the advantages of the SZI$_{snow}$.

---

## Author Comment (AC4)

Response to Referee #3:

The authors would like to thank you for the constructive and thoughtful comments. We have addressed all your suggestions, leading to a much improved and complete manuscript. The following comments are addressed in the sequence as they were asked. We also respond to each point and clarify the corresponding changes adopted in the revised manuscript. The original comments are copied from the report with a **bolded font in black**, and our answers are in blue. Manuscript changes are in ***bold italic***.

Thank you again for your time and effort in reviewing our manuscript.

Yours sincerely

Baoqing Zhang (on behalf of all co-authors)

**Reviewer #3**

Tian et al. proposed a drought dataset for indicating drought across multiple categories and temporal scales. The proposed SZIsnow dataset includes different physical water-energy processes, especially snow processes. The evaluation for different spatiotemporal scales indicates the dataset can distinguish different types of drought. The SZIsnow shows superior performance over the cold regions. In addition, the dataset successfully described large-scale drought events over the world.

The purpose of the proposed work is clear and essential in order to establish a drought dataset including snow information for other studies related to drought. The manuscript is generally well-structured, the method description, evaluation, and data availability were well-written. However, several points should be addressed in the revised manuscript. I offer comments below in the hope this can be used to improve the paper further.

Reply: We appreciate the reviewer's comments on our work.

**Major comments**

First, in lines 102-103, the author stated that the GLDAS-2 drives the Noah land surface model (LSM), forced by the global Princeton meteorological forcing data, to approximate the observed land surface state. However, the Noah is not the only land surface model used by the GLDAS-2. I get the information from the URL: https://ldas.gsfc.nasa.gov/gldas. Could you explain why you chose the Noah model? Are there any differences among these land surface models?

Reply: We thank the reviewer for this excellent comment. At present, there are two versions of the GLDAS product: GLDAS-1 and GLDAS-2. The GLDAS-1 drives four land surface models (LSMs) at a spatial resolution of 1.0° from 1979 to 2017, including the Community Land Model (CLM), Mosaic model (MOS), NOAH model (NOAH1.0),

and Variable Infiltration Capacity model (VIC). On the other hand, the GLDAS-2 only drives the upgraded version of Noah model (NOAH2.0) at a spatial resolution of 0.25°from 1948 to 2010. Here we chose 18 snow-influenced basins from the 32 global basins to discern the differences in the snow processes representation among the five LSMs mentioned above.

We compared and analyzed snow processes of the LSMs in terms of snowfall (Figure R1), snowmelt (Figure R2), and the snow water equivalent (SWE, Figure R3). As we can see, there are large differences for the snowmelt processes related to model snow physics, surface meteorological forcing (e.g., total precipitation, rainfall, snowfall, 2-meter air temperature), and sublimation in GLDAS-1 and GLDAS-2. As the same forcing is used for the four models in GLDAS-1, its snowfall is very similar, although CLM has much more snowfall than the other three models. The reason is that all other three models use 0 °C 2-meter air temperature to separate total precipitation into rainfall and snowfall. However, CLM uses the Jordan algorithm (Jordan 1991) to separate total precipitation into rainfall and snowfall. The Jordan algorithm accounted for frozen rain, frozen rain and snow mix, and snowfall uses 2.5 °C 2-m air temperature to separate more total precipitation algorithm into snowfall. Additionally, small snowmelt values in GLDAS-1 Noah1.0 is due to larger sublimation. In general, compared GLDAS-2 with GLDAS-1, major SWE differences come from surface meteorological forcing data (precipitation and air temperature), although model snow physics also plays an important role.

We chose the GLDAS-2 as the forcing data to derive the $SZI_{snow}$, since our survey of the previous studies shows that the GLDAS-2 has a better performance than the GLDAS-1. Although the GLDAS-2 products have relatively large uncertainties in snow processes over regions with low quality meteorological forcing data and uncertain model physical processes, hydrological and meteorological studies over areas that lack complete set observations would benefit from GLDAS-2 products. This poses a dilemma. Nevertheless, using GLDAS-2 products to represent the land surface water-energy states and fluxes for large-scale hydrological or meteorological research is currently one of the commonly used methods, especially over regions lacking highquality *in-situ* observations. Despite uncertainties and errors in snow simulation of the GLDAS-2 product, the accuracy of the input water budgets does not influence the conceptual and technical improvement of the $SZI_{snow}$ by considering the impact of snow dynamics on water supply and demand in drought characterization.

*Jordan, R., 1991: A one-dimensional temperature model for a snow cover: Technical documentation for SNTERERM.89. Special Rep. 91–16, Cold Region Research and Engineers Laboratory, U.S. Army Corps of Engineers, Hanover, NH, 61 pp.*

[Figure]

**Figure R1.** Monthly snowfall of different LSMs at 18 snow-influenced basins during 1979-2010.

[Figure]

**Figure R2.** Monthly snowmelt of different LSMs at 18 snow-influenced basins during 1979-2010.

[Figure]

**Figure R3.** The monthly SWE of different LSMs in GLDAS at 18 snow-influenced basins during 1979-2010.

Second, the author adopted the log-logistic distribution to standardize precipitation, streamflow, and soil water storage to compute the Standardized Precipitation Index, Standardized Streamflow Index, and Standardized Water Storage Index. As I know, other probability distributions can be used to standardize. Are your evaluation results independent of different methods?

Reply: We have tested the validity of four applied distribution functions for precipitation (P), D (D = P − PET), WER, Z, and $Z_{snow}$ across different climate zones. These climate zones were classified on the basis of the Aridity Index (AI), which is computed by the ratio of the long-term mean annual PET to P (AI = PET/P). A drier local climate condition corresponds to a higher AI value. Therefore, regions with AI

equal to 0.76, 1.45, 2.5, 4.4, and 79.9 represent humid, sub-humid, semi-arid, arid, and extreme arid regions.

As shown in Figure R4, we tested four possible three-parameter distributions to model the P, D, WER, Z, and $Z_{snow}$ values at different climate zones, including Pearson III, log-logistic, lognormal, and general extreme value (GEV) distribution. For this purpose, the L-moment ratio diagram was used to examine the performance of different distributions because it allows comparing the empirical frequency distribution of P, D, WER, Z, and $Z_{snow}$ time series at different climate regions with several theoretical distributions. The results show the log-logistic distribution generally fits the P, D, WER, Z, and $Z_{snow}$ data the best across multiple climate zones. Therefore, the log-logistic distribution can reasonably be adopted for standardizing the P, D, WER, Z, and $Z_{snow}$ data in calculating the SPI, SPEI, SWI, SZI, and $SZI_{snow}$.

[Figure]

[Figure]

**Figure R4.** Empirical and modeled values (f(x) and F(x) or probability density function and cumulative distribution function, respectively) using the Pearson III, log-logistic, lognormal, and general extreme values (Gen. Ext. Value) distributions of the P, D, WER, Z, and Z$_{snow}$ series at monthly scale for deriving the SPI, SPEI, SWI, SZI, and SZI$_{snow}$.

**Minor comments**

**1) Lines 19-20, 22**

**Some numbers keep two decimals, and some do not.**

Reply: The number in line 22 will be revised and consistent with other numbers. In the revised manuscript, this line will be changed to "*of which 68.38% had a duration longer than 6 months.*"

**2) Lines 99-101**

**Show some reasons to explain why the better performance of GLDAS-2 compared to that of GLDAS-1?**

Reply: Various studies have shown that the GLDAS-2 has a better performance than the GLDAS-1. These studies pointed out several causes for such better performance of GLDAS-2, and here we summarized the causes as follows. GLDAS-1 has serious discontinuity issues in its forcing data. Especially, GLDAS-1 precipitation data have larger errors in 1996, and the snowfall amount has approximately doubled after 2000. These discontinuity issues are primarily attributed to several switches of the forcing data of GLDAS-1. In contrast, GLDAS-2 precipitation has much better temporal continuity than GLDAS-1 precipitation, as GLDAS-2 uses the bias-corrected Princeton meteorological forcing dataset. Additionally, evaluations over the high-altitude regions indicate that GLDAS-2 models perform better than the GLDAS-1 models in simulating runoff. This is mainly because the GLDAS-2 models integrate glacier melt runoff in the simulation, whereas the GLDAS-1 models demonstrate no glacier melt runoff.

Following the comment, we will add these causes in the revised manuscript to explain the better performance of GLDAS-2 compared to GLDAS-1 as follows:

*This is mainly attributed to that GLDAS-1 has serious discontinuity problems in its meteorological forcing dataset due to switches in its forcing data. In contrast,*

*GLDAS-2 has a better temporal continuity, using the bias-corrected Princeton meteorological forcing dataset. Additionally, evaluations for the high-altitude regions indicate that GLDAS-2 performs better in streamflow simulation because GLDAS-2 considers streamflow from glacier melt in its simulation, but the GLDAS-1 did not.*

**3) Line 140**

**I did not quite understand why the WER is regarded as a comprehensive drought indicator? What is a comprehensive drought indicator? Is there any definition for it? Please give more information.**

Reply: The residual water–energy ratio (WER) was first suggested by Liu et al. (2017). As mentioned in the manuscript, the ratio of the residual available water to the residual energy (PET−ET) is relatively low (large) during drought (wet) events relative to normal conditions. Defining this ratio as WER = (P − ET)/(PET − ET), Liu et al. (2017) proposed a method for examining the response of the surface water-energy fluxes to drought based on WER.

Droughts are generally classified into three categories (meteorological, hydrological, and agricultural droughts) based on their physical characteristics. Nevertheless, these different categories make it difficult to objectively quantify drought features. Moreover, different types of droughts are considered to be interchangeable. Drought can convert from one type to another as it evolves in time and space. Zhang et al. (2019) thus recommend employing the WER, a comprehensive drought indicator, to comprehensively describe drought through a water-energy balance perspective. We can objectively and easily quantify drought across different spatial scales through such a perspective.

**4) Line 155**

**Did you evaluate the capacity of your dataset across different climate zones? I did not see these mentioned climate zones in the following sections of the manuscript.**

**I guess you wanted to say that the evaluation was conducted over different geographical parts of the world. Please clarify this.**

Reply: This is a typo. This line will be corrected in the revised manuscript as "*multiscalar drought across different geographical parts of the world*."

**5) Lines 221-222**

**What is the name of dimensions in your 3D and 2D dataset? Please clarify this.**

Reply: The names of dimensions in the 3D dataset are month, latitude, and longitude. The names of dimensions in the 2D dataset are latitude and longitude. This sentence will be modified in the revised manuscript as follows:

*The SAD method firstly uses a monthly three-dimensional (3D, $month \times latitude \times longitude$) gridded drought index dataset to identify two-dimensional (2D, $latitude \times longitude$) drought clusters in each time step.*

**6) Line 273**

**It seems this content has been repeated in the main text. Delete it to make the caption more concise.**

Reply: This content will be removed from the revised manuscript following the comment.

**7) Line 353**

**Is the geographical extent of Oceania equal to that of Australia? Give an exact definition.**

Reply: The geographical extent of Oceania is not equal to Australia. This study defines Oceania as Australia, New Zealand, Papua New Guinea, and the Pacific Islands. This content will be added in the revised manuscript as follows:

*Oceania is defined as Australia, New Zealand, Papua New Guinea, and the Pacific Islands.*

**8) Why wasn't Greenland included in Figure 6? It seems all the Greenland are missing values.**

Reply: Greenland was excluded from our study because about 80% of its surface is ice-capped. In addition, Greenland has a small population of nearly 56,100 (in 2016) on an area of 2,166,086 km², making Greenland the least densely populated place on earth. Thus, the influence of droughts on this island is minimal, and the droughts in Greenland were not of interest for this study.

Following the comment, the reason for the exclusion of Greenland will be added in the revised manuscript as follows:

*Our study excluded Greenland due to its sizeable ice-capped area (about 80% of the island) and low-density population.*

**9) When you did the SAD analysis, how to process the drought over the Sahara Desert?**

Reply: When we did the SAD analysis, drought events were tracked through time by searching for overlapping grid cells between clusters at contiguous time steps. Clusters were allowed to propagate into the Sahara until their centroids fell within the specified domain that is outside the Sahara. Droughts whose centroids fell within the Sahara Desert (20°N–25°N, 17°W–34°E) were removed from our analysis since droughts in these regions were not of interest for this study.

Following the comment, how to process droughts over the Sahara in the SAD analysis will be added in the revised manuscript as follows:

*Droughts in the Sahara Desert were not concerned in our study. During the SAD analysis, clusters were allowed to propagate into the Sahara (20°N–25°N, 17°W–*

*34°E), and these clusters would be retained if their centroids fell outside the Sahara Desert. In contrast, drought clusters were discarded if their centroids locate in the Sahara Desert.*

**Technical comments:**
**1) Figure 1: The font size of the description to compare the strength and weakness of each index is small.**

Reply: In the revised manuscript, we will enlarge the font size of the description to compare the strength and weakness in Figure 1.

**2) Figure 4: The font should be the same, use one type.**

Reply: We will revise the font and use the same font in Figure 4 in the revised manuscript.

**3) Figure 6: The stippling seems unclear.**

Reply: We will increase the resolution of Figure 6 in the revised manuscript.

**4) Figure 9: Please supply information on the geographic coordinate system used in this figure. It can help others to compare their results with yours.**

Reply: Figure 9 uses the geographic coordinate system (latitude and longitude). We will add this information into the caption of Figure 9 in the revised manuscript.

**5) Figure S2: Adjust the minimum value of the legend. There is no grid with a SZIsnow value less than -4.0.**

Reply: The minimum level of the SZIsnow was set as -4.0 in Figure S2, since we wanted to keep the legend of Figure S2 to be consistent with that in Figure 9.

**Comments for the dataset**

**1) Table 1 in the metadata file should be kept the same as Table 2 in the manuscript.**

Reply: We will replace Table 1 in the metadata file with Table 2 in the manuscript.

**2) Add information relative to the size of decompressed files in the metadata file.**

Reply: Following the comment, we will add the below information relative to the size of decompressed files in the metadata file:

*The size of each zipped file is 14.5 GB (gigabyte), and each file in a zipped file is 2.43 GB. The total size of all the decompressed files is 116 GB.*

**3) I recommend providing a thumbnail of your dataset in the metadata file.**

Reply: Following the comment, we will add a thumbnail of our dataset in the metadata file as follows:

[Figure]

**Figure R5.** Spatial distribution of the SZI$_{snow}$ on July 1992 at a 6-month timescale.

**4) If possible, provide some scripts for potential readers to plot your data.**

Reply: Following the comment, we will provide two scripts for potential readers to plot the $SZI_{snow}$ data in the metadata file. The first script is coded based on Python programming language, and another is coded based on the NCAR Command Language (NCL).

---

## Author Response (AR1)

"A global drought dataset of standardized moisture anomaly index incorporating snow dynamics (SZI$_{snow}$) from 1948 to 2010" by Tian et al.

Response to Referees

We thank the referees for their time and effort in reviewing our manuscript. We are also grateful for the constructive and thoughtful comments that have substantially improved the quality of this manuscript. Here we address each of the referee's comments and clarify the corresponding changes in the revised manuscript. The original comments are in **black bolded font**, our responses are in blue, and manuscript changes are in ***bold italic***.

**Referee #1:** https://doi.org/10.5194/essd-2021-399-RC1

**This manuscript introduces a drought product with a multitype and multiscalar drought index, SZI$_{snow}$. The SZI$_{snow}$ dataset considers a relatively comprehensive extent of the hydrometeorological variables associated with drought development. It uniquely incorporates snow processes in the derivation of a drought index. This consideration is important as global warming has been affecting hydrological processes over the snow-covered regions. The dataset was evaluated across different spatial scales. The result shows the SZI$_{snow}$ has good performance over the snow-covered regions. The dataset was also used to survey the evolution of large drought events with the severity-area-duration method.**

**The topic of the study is interesting and well fits the scope of the journal, especially for this special issue. The manuscript is well written, logically organized, and the details of the derivation of the SZI$_{snow}$ are easy to follow. The data processing is careful and well documented, and the dataset was friendly to access. However, there are still some concerns that need to be addressed. Thus, I am supportive of**

**the publication after a minor revision to further improve the quality or make it more clear for the readers to understand the results. Below are my suggestions:**

Reply: We appreciate the referee's positive comments and interest in our work. Further details are provided in response to the specific comments below.

**General comments**

**#1 The main improvement in the SZI$_{snow}$ is the consideration of snow related processes, thus it is reasonable that the SZI$_{snow}$ in the snow-covered basins have a better performance than that of SZI as shown in Figure 3. However, I found that both the SZI$_{snow}$ and SZI have a similar performance over the snow-free basins. Please clarify this similar performance.**

Reply: We agree with this comment that the SZI$_{snow}$ has a similar performance with the SZI over the snow-free basins. Such similar performance is mainly attributed to the fact that P$_{snow}$, PSM, and PSA values are close to zero over basins at low-latitude and low-altitude snow-free areas, and thus the calculation of SZI$_{snow}$ converges to the snow-free basins. Therefore, the performance of SZI$_{snow}$ and SZI are consistent with each other over snow-free areas.

For the SZI$_{snow}$:

$$\begin{cases} P = P_{rainfall} + P_{snowfall} \\ Z_{snow} = P - \hat{P}_{snow} \\ \hat{P}_{snow} = \alpha_j PET + \beta_j PR + \gamma_j PRO + \delta_j PSA - \varepsilon_j PL - \varphi_j PSM \end{cases}$$

For the SZI:

$$\begin{cases} Z = P_{rainfall} - \hat{P} \\ \hat{P} = \alpha_j PET + \beta_j PR + \gamma_j PRO - \delta_j PL \end{cases}$$

Following the comments, we have added some information, which explains why the performance of the SZI$_{snow}$ is similar to that of the SZI in snow-free areas, into Section 4.1.1 of the revised manuscript (Page 14, Lines 296-300) as follows:

*Besides the outperformance of the SZI$_{snow}$, it should be noted that the SZI$_{snow}$ has a similar performance with SZI over the snow-free basins. Such similar performance is mainly owing to the fact that the values of P$_{snow}$, PSM, and PSA are close to zero*

*over basins at low-latitude and low-altitude (snow-free) areas, leading to the calculation of $SZI_{snow}$ converges to the snow-free basins. Therefore, the performance of $SZI_{snow}$ and SZI are consistent with each other over snow-free areas.*

**#2 The evaluation of the $SZI_{snow}$ is important compared to current drought indices (for example the scPDSI). Besides the SPI, I think the scPDSI is also a good index to assess the meteorological drought. Thus, it is necessary to evaluate the performance of $SZI_{snow}$ to capture meteorological drought compared with other indices instead of only with SPI. This can help to confirm the robustness of your conclusion.**

Reply: Following the comments, in addition to the SPI, we adopted two other mainstream drought indices (SPEI and scPDSI) to compare their performance in monitoring meteorological drought. Figure R1 shows the responses of multiple drought indices to meteorological drought. As we can see, the performance of $SZI_{snow}$ is prominent and superior to SZI, SPEI, and scPDSI in identifying meteorological drought at multiple temporal scales. This further confirms our prior conclusion.

Following the comments, we have added the following information into Section 4.1.1 of the revised manuscript (Page 13, Lines 275-278), and Figure R1 has been added as Figure S3 into the supplementary material:

*In addition to the SPI, we adopted two other mainstream drought indices (SPEI and scPDSI) to compare their performance in monitoring meteorological drought. As shown in Fig. S3, the performance of $SZI_{snow}$ is prominent and superior to SZI, SPEI, and scPDSI in identifying meteorological drought at multiple temporal scales. The selection of reference drought indices did not influence the reliability of our conclusion.*

[Figure]

Figure R1. The Pearson correlation coefficients between SPEI and SPI (i.e., SPEI-SPI), scPDSI-SPI, SZI-SPI, and SZI$_{snow}$-SPI at 1- to 48-month time scales in the selected 32 large basins during 1948–2010. The dominant climate regime of each basin is shown in the parentheses.

**Specific comments**

**#1 Is it possible to include the acronym of your new drought index in the title? Such inclusion can enhance the recognizability of your dataset and facilitate others to cite and employ your dataset.**

Reply: The title has been changed to "*A global drought dataset of standardized moisture anomaly index incorporating snow dynamics (SZI$_{snow}$) from 1948 to 2010*" in the revised manuscript.

**#2 I suggest adding the spatial resolution of your dataset in the Abstract section.**

Reply: The spatial resolution of a dataset is essential for the Earth System Science Data journal. The spatial resolution of our proposed dataset is 0.25 degrees. This content has been added in the revised manuscript as follows (Page 1, Line 12):

*Here, we present a global monthly drought dataset with a spatial resolution of 0.25° from 1948 to 2010 based on a multitype and multiscalar drought index, the standardized moisture anomaly index incorporating snow dynamics (SZI$_{snow}$), driven by systematic fields from an advanced data assimilation system.*

**#3 The author listed various current drought indices in Figure 1. Please make sure the corresponding references of these indices are supplied in the manuscript.**

Reply: Some drought indices in the first row of Figure 1 do not have corresponding references because those indices (e.g., SAI and PHDI) are not mentioned in the main text of the manuscript. Thus, we have added references of these non-mentioned indices in the supplementary material.

**#4 Line 38: Correct the "focus" to "focuses".**

Reply: This grammar problem has been corrected in the revised manuscript (Page 2, Line 39). Such problems have been fixed throughout the revised manuscript.

**#5 Line 288: Did the two subplots in Figure S3 have identical contour levels and color bar? If did, please remove one color bar.**

Reply: Yes, they did. We have removed one color bar of this figure, and the revised figure has been updated as Figure S4 in the revised supplementary material as follows:

[Figure]

*Figure S4. Comparison between the SZI and SZI$_{snow}$ in the context of their performance over the Arctic region. (a) Spatial distribution of the correlation coefficients of the SZI$_{snow}$–SWI over a 12-month timescale. (b) Spatial distribution of the correlation coefficients of the SZI–SWI over a 12-month timescale.*

**#6 Line 144: Consider adding a comma after the introductory phrase "meanwhile".**

Reply: This grammar problem has been corrected in the revised manuscript (Page 6, Line 153).

**#7 Line 300: What does the shading in the Figure 4d panel mean? It should be clarified in the Figure caption. The inset of the Figure 4d panel seems not clear to me, you can increase the resolution of your figure.**

Reply: The shading in this panel denotes the range of correlation coefficients of the SZI$_{snow}$–SWI and SZI–SWI. The upper (lower) boundary is the maximum (minimum) value. In addition, the original resolution of this figure is 300 dpi. We have increased its resolution to 600 dpi, and this figure has been updated as Figure 5 in the revised

manuscript. An explanation of the shading has been added into the caption of this figure. as follows (Page 17, Lines 339-340):

*The shading denotes the range of correlation coefficients. The upper (lower) boundary is the maximum (minimum) value.*

**#8 Line 315: Did the subplots in the left and middle columns of Figure 5 have identical contour levels and color bar? If did, please remove one color bar.**

Reply: We corrected this problem of this figure, and updated it as Figure 6 in the revised manuscript as follows (Page 18, Line 350):

[Figure]

*Figure 6: Spatial distribution of correlation coefficients of the $SZI_{snow}$–SWI (left column) and those of the SZI–SWI (middle column), and the differences between the two (right column = left column minus middle column) over the Tibetan Plateau at different timescales (6, 9, 12, and 15 months).*

**#9 Line 330: Some corner strings (e.g. "a", "b" in panels) are bolded, but some aren't. Please keep the format of these strings consistent across the manuscript.**

Reply: We checked and modified the appearance of corner strings throughout the manuscript. As a result, the format of these strings is now consistent across the revised manuscript.

**#10 I recommend adding the values of trends in Figure 7 so that the reader can know more information from the figure.**

Reply: We have added the values of trends in this figure and updated it as Figure 8 in the revised manuscript (Page 20, Line 378) as follows:

[Figure]

*Figure 8: Time series of (a) global dry land area (% yr$^{-1}$) and (b) $Z_{snow}$ (mm yr$^{-1}$) between 1948 and 2010. The dry land area was calculated based on the $SZI_{snow}$ at a 12-month timescale. The dashed lines denote the linear trends, and the numbers represent the value of change rate.*

**#11 Line 352: It is a good way to use a country to describe the area of a drought event. Please add a number to show how large is the size of Guatemala.**

Reply: The area of Guatemala is 108,889 km² that has been added in the revised manuscript as follows (Page 21, Lines 389-390):

*The most spatially extensive drought occurred over Asia in August 2008 (drought lasted from November 2007 to June 2009) and covered an area of approximately 11 million km² (roughly 100 times Guatemala's national territory area of 108,889 km²).*

**#12 Line 400: The Interdecadal Pacific Oscillation just appears one time, it is not necessary to provide an abbreviation for it. Please correct the same problems throughout the manuscript.**

Reply: The abbreviation of the Interdecadal Pacific Oscillation (i.e., IPO) has been removed in the revised manuscript (Page 24, Line 438).

**Dataset**

**I downloaded the compressed files including all the files from http://doi.org/10.5281/zenodo.5627369. With the software of "ncdump" and "Panoply", I checked the data files and had no problems to read and visualize the data. All the data is consistent as they were described in the manuscript. Here I give some recommendations to improve the user-friendliness of the proposed dataset.**

**#1 The compressed file with a suffix of ".zip". Is it possible add an introduction about how to unzip this kind of file?**

Reply: Our data were compressed by a free and open-source file archiver, 7-Zip, due to its higher compression ratio than other common software. Following the comment, we will add below introduction about how to unzip this kind of file in the Metadata file at the National Tibetan Plateau/Third Pole Environment Data Center (http://data.tpdc.ac.cn/en/disallow/b039fde6-face-4d24-af45-d238a6af18b7/):

*All the files were compressed by the 7-Zip software. The 7-Zip is a free and open-source file archiver and can be downloaded from the URL: https://www.7-zip.org/download.html. After opening the URL, you can choose a suitable install file based on your operating system. The operation of 7-Zip is similar to the standard file archiver.*

**#2 The SZIsnow datasets have different timescales, thus, months before the timescale are set as missing values. It would be nice to give a clear introduction about the missing values in your dataset.**

Reply: Following the comment, we will add below introduction about the missing values of the $SZI_{snow}$ dataset in the Metadata file at the National Tibetan Plateau/Third Pole Environment Data Center (http://data.tpdc.ac.cn/en/disallow/b039fde6-face-4d24-af45-d238a6af18b7/):

*The $SZI_{snow}$ is a multiscalar drought index, and the $SZI_{snow}$ variable in the data file has different timescales. Thus, for a $SZI_{snow}$ variable (len, time, lat, lon) at n months timescale, the first n-1 months (from 1 to n-1) in the SZIsnow variable are missing values.*

**Referee #2:** https://doi.org/10.5194/essd-2021-399-RC2

**The authors developed a global dataset of standardized drought indicators incorporating snow information. They first evaluated the indicator and then employed it for the drought temporal-spatial analysis across the globe. This is an interesting dataset for the drought analysis at the global scale. I have several comments as follows.**

Reply: We appreciate the referee's comments and interest in our work.

**General comments**

**Figure 1 In the box of hydrological drought. The advantage (green box) is "considers snowpack and water storage" and the disadvantage (red box) is "unsatisfactory performance over snow covered regions". What is the reason for the "unsatisfactory performance over snow…" if these indices already consider "snow pack"?**

Reply: We would like to apologize for this vague description. We listed three drought indices in the box of hydrological drought (i.e., SDI, PHDI, and SWI). The SDI only requires monthly streamflow for its calculation. The PHDI requires monthly temperature, precipitation, and water-holding capacity of soils for its calculation. Thus, the SDI and PHDI have unsatisfactory performance over snow-covered regions because the input parameters of the two indices (SDI and PHDI) do not include parameters related to snow.

In the three listed indices in the box of hydrological drought, only the SWI considers snowpack in its calculation. Its input parameters include streamflow, precipitation, reservoir storage, and snowpack. Nevertheless, the SWI is basin-dependent by calculating at the basin level, and it is difficult to compare basins, not to mention at the grid level.

To clarify this vague description, we have updated Table S1 in the supplementary material, which clearly lists the strengths and weaknesses of the drought index not mentioned in the main text of the manuscript.

**Figure 1 It seems the SZI addresses all the challenges of the indices mentioned in this figure. Is there any remaining disadvantage or limitation of SZI? Please clarify.**

Reply: We agree with this comment and admit that the SZI can not address all the challenges of the indices mentioned in this figure. The SZI does still have some remaining disadvantages or limitations. Figure 1 is used to show our motivation and train of thought for the development path of the index. We thus did not put too much emphasis on the limitations of the SZI in the original manuscript. In general, there are two main limitations of the SZI. The first one is that its computation is more complex and difficult than the SPI or SPEI. Another limitation is that its calculation requires long-term climatic and hydrologic records, making it unsuitable for short-term drought identification and monitoring.

Following the comments, we will add the following information into the fourth paragraph of Introduction section in the revised manuscript (Page 3, Lines 65-67):

*Additionally, there are two main limitations of the SZI. The first one is that its computation is more difficult than the SPI or SPEI. Another limitation is that it needs a long-term serial of hydrometeorological records, making it unsuitable for short-term drought studies.*

**Lines 133: "We evaluated the ability of the SZIsnow and SZI to capture different" It is generally hard for a single drought indicator to capture all types of droughts. Is the SZI designed to capture all drought types?**

Reply: We cannot agree more with this comment. It is generally hard for a single drought indicator to capture all types of droughts. Indeed, it is essential to realize this challenging objective. This is because the impacts of droughts are significant and widespread, affecting many economic sectors and people at any one time. In addition, different types of droughts are considered to be interchangeable. Drought can convert from one type to another as it evolves in time and space. Nevertheless, different administrative departments nowadays employ various drought indices for drought

management, leading to unaligned action plans against drought. Without alignment, there is likely to be considerable delay in action at the onset of drought in an area or region. Therefore, our study strives to design a drought index, SZI, to capture all drought types. Though it is challenging to develop a multitype index and our proposed index is not 100-percent perfect, our work is on the right track. In this way, different sectors of society can collaborate to synergistically fight against drought using a comprehensive drought index.

**Lines 201-202: Here P and Psnow are used to define the water supply deficit. This equation is only applied in regions and seasons with snowfall, right? What about other regions (e.g., tropics)? Do you use a different set of equations to calculate SZI?**

Reply: Our equation can be applied in regions and seasons with and without snowfall. For the regions without snowfall (e.g., tropics), the items relevant to snow in the equations are set to zero for the calculation of SZI$_{snow}$. For example, in the following equation, the $P_{snowfall}$, $\delta_j PSA$, and $\varphi_j PSM$ are set to zero when they are used for situations without snowfall.

$$
\begin{cases}
P = P_{rainfall} + P_{snowfall} \\
Z_{snow} = P - \hat{P}_{snow} \\
\hat{P}_{snow} = \alpha_j PET + \beta_j PR + \gamma_j PRO + \delta_j PSA - \varepsilon_j PL - \varphi_j PSM
\end{cases}
$$

This comment is an important reminder for our study. Following the comments, we have added some information into Section 3.1.3 to explain how to calculate SZI$_{snow}$ for regions and seasons without snowfall in the revised manuscript (Page 11, Lines 215-218):

*In addition, our equations can be applied in regions and seasons without snowfall. For regions without snowfall (e.g., tropics), the items relevant to snow in the equations of Table 1 are set to zero for the calculation of SZI$_{Isnow}$. For example,*

$\delta_j PSA$, $\varphi_j PSM$, and $P_{snowfall}$ *are set to zero when they are used for situations without snowfall.*

**Lines 244-245: Here the authors used SPI to evaluate the proposed index SZIsnow. SPI mainly reflects precipitation-related droughts. There may not be snow information in SPI (it does not incorporate snowfall, right?). How do we know a higher SZIsnow-SPI correlation reflects better performance of the proposed index? Please justify this.**

Reply: The standardized precipitation index (SPI) uses historical precipitation records for any location to develop a probability of precipitation that can be computed at any number of timescales, from 1 month to 48 months or longer. The only input parameter of SPI is precipitation, which is the greatest strength of SPI. Precipitation is regarded as the sum of rainfall and snowfall during the calculation of SPI. Thus, the SPI considers the snow information, and it is reasonable to use SPI for the evaluation of the $SZI_{snow}$.

**Lines 294-295: It seems the proposed index performs better for long-time scales. Any specific reason for this? Please clarify.**

Reply: We agree with this comment that the proposed index performs better for long-time scales. There are mainly three reasons for its better performances at long timescales. Firstly, the magnitude of the accuracy of the moisture anomaly $Z_{snow}$ (Equation 9 of Table 1) is small at a short timescale (e.g., 1-month). In comparison, the accuracy of $Z_{snow}$ becomes relatively larger at a long timescale (e.g., 12-month) because the 12-month $Z_{snow}$ on a certain date is an accumulative value by summing up monthly $Z_{snow}$ of the prior 12 months. Accordingly, the improvement is more evident at a long timescale. Secondly, it usually takes a long time from the arrival of precipitation (rainfall and snow) to precipitation becomes different forms of useable water. Precipitation infiltrates into the groundwater takes longer than it directly converts to overland flow. In addition, the snowpack accumulates in the cold season, and snowmelt drains into the soil or directly into the river channel in the warm season, which leads to a several-month to 1-year lag response in the soil wetness and total water storage

variability. At last, long-term droughts always cause more widespread and severe consequences than short-term droughts. The capacity of the $SZI_{snow}$ to identify and monitor long-term drought is the focus of our study.

**Figure 6. This trend analysis for each grid is performed for the annual time series? Or for a season? Please clarify.**

Reply: We performed the trend analysis for each grid in this figure for the annual time series. Following the comments, we have provided a clearer caption (Page 19, Lines 368-369) and description (Page 19, Line 356) for this figure (has been updated as Figure 7) in the revised manuscript as follows:

*Figure 7: Spatial distribution of the linear annual trend (changes per 50 years) in the SZI$_{snow}$ during the period 1948–2010, at various timescales. The stippling denotes the trend being statistically significant at the 95% confidence level.*

*The spatial distribution of the linear annual trend in the SZI$_{snow}$ over different timescales (i.e., 3-, 6-, 12-, 15-months) is shown in Fig. 7.*

**Figure 7 Any speculation for the low SZI and high dry areas during 1985-1990?**

Reply: The $SZI_{snow}$ employs precipitation amount (referred as $\widehat{P}_{snow}$ ) that is climatically appropriate for existing conditions to quantify the regional water demand. The moisture anomaly $Z_{snow}$ is defined as the difference between the actual Precipitation $(P)$ and $\widehat{P}_{snow}$, which is an appropriate indicator for regional water deficiency or surplus. Figure 7 in the original manuscript has been updated as Figure 8 (Page 20, Line 378) in the revised manuscript. In the bottom subfigure, we used $Z_{snow}$, instead of $SZI_{snow}$, to represent dry and wet conditions over a region. There is a negative correlation between $Z_{snow}$ and dry areas. Thus, the high dry areas correspond with the low $Z_{snow}$ during 1985-1990.

**Referee #3:** https://doi.org/10.5194/essd-2021-399-RC3

**Tian et al. proposed a drought dataset for indicating drought across multiple categories and temporal scales. The proposed SZIsnow dataset includes different physical water-energy processes, especially snow processes. The evaluation for different spatiotemporal scales indicates the dataset can distinguish different types of drought. The SZIsnow shows superior performance over the cold regions. In addition, the dataset successfully described large-scale drought events over the world.**

**The purpose of the proposed work is clear and essential in order to establish a drought dataset including snow information for other studies related to drought. The manuscript is generally well-structured, the method description, evaluation, and data availability were well-written. However, several points should be addressed in the revised manuscript. I offer comments below in the hope this can be used to improve the paper further.**

Reply: We appreciate the referee's comments on our work.

**Major comments**

**First, in lines 102-103, the author stated that the GLDAS-2 drives the Noah land surface model (LSM), forced by the global Princeton meteorological forcing data, to approximate the observed land surface state. However, the Noah is not the only land surface model used by the GLDAS-2. I get the information from the URL: https://ldas.gsfc.nasa.gov/gldas. Could you explain why you chose the Noah model? Are there any differences among these land surface models?**

Reply: We thank the referee for this excellent comment. At present, there are two versions of the GLDAS product: GLDAS-1 and GLDAS-2. The GLDAS-1 drives four land surface models (LSMs) at a spatial resolution of 1.0° from 1979 to 2017, including the Community Land Model (CLM), Mosaic model (MOS), NOAH model (NOAH1.0), and Variable Infiltration Capacity model (VIC). On the other hand, the GLDAS-2 only

drives the upgraded version of Noah model (NOAH2.0) at a spatial resolution of 0.25°from 1948 to 2010. Here we chose 18 snow-influenced basins from the 32 global basins to discern the differences in the snow processes representation among the five LSMs mentioned above.

We compared and analyzed snow processes of the LSMs in terms of snowfall (Figure R2), snowmelt (Figure R3), and the snow water equivalent (SWE, Figure R4). As we can see, there are large differences for the snowmelt processes related to model snow physics, surface meteorological forcing (e.g., total precipitation, rainfall, snowfall, 2-meter air temperature), and sublimation in GLDAS-1 and GLDAS-2. As the same forcing is used for the four models in GLDAS-1, its snowfall is very similar, although CLM has much more snowfall than the other three models. The reason is that all other three models use 0 °C 2-meter air temperature to separate total precipitation into rainfall and snowfall. However, CLM uses the Jordan algorithm (Jordan 1991) to separate total precipitation into rainfall and snowfall. The Jordan algorithm accounted for frozen rain, frozen rain and snow mix, and snowfall uses 2.5 °C 2-m air temperature to separate more total precipitation algorithm into snowfall. Additionally, small snowmelt values in GLDAS-1 Noah1.0 is due to larger sublimation. In general, compared GLDAS-2 with GLDAS-1, major SWE differences come from surface meteorological forcing data (precipitation and air temperature), although model snow physics also plays an important role.

We chose the GLDAS-2 as the forcing data to derive the $SZI_{snow}$, since our survey of the previous studies shows that the GLDAS-2 has a better performance than the GLDAS-1. Although the GLDAS-2 products have relatively large uncertainties in snow processes over regions with low quality meteorological forcing data and uncertain model physical processes, hydrological and meteorological studies over areas that lack complete set observations would benefit from GLDAS-2 products. This poses a dilemma. Nevertheless, using GLDAS-2 products to represent the land surface water-energy states and fluxes for large-scale hydrological or meteorological research is currently one of the commonly used methods, especially over regions lacking highquality *in-situ* observations. Despite uncertainties and errors in snow simulation of the GLDAS-2 product, the accuracy of the input water budgets does not influence the conceptual and technical improvement of the $SZI_{snow}$ by considering the impact of snow dynamics on water supply and demand in drought characterization.

Jordan, R., 1991: A one-dimensional temperature model for a snow cover: *Technical documentation for SNTERERM.89*. Special Rep. 91–16, Cold Region Research and Engineers Laboratory, U.S. Army Corps of Engineers, Hanover, NH, 61 pp.

[Figure]

Figure R2. Monthly snowfall of different LSMs at 18 snow-influenced basins during 1979-2010.

[Figure]

Figure R3. Monthly snowmelt of different LSMs at 18 snow-influenced basins during 1979-2010.

[Figure]

Figure R4. The monthly SWE of different LSMs in GLDAS at 18 snow-influenced basins during 1979-2010.

**Second, the author adopted the log-logistic distribution to standardize precipitation, streamflow, and soil water storage to compute the Standardized Precipitation Index, Standardized Streamflow Index, and Standardized Water Storage Index. As I know, other probability distributions can be used to standardize. Are your evaluation results independent of different methods?**

Reply: We have tested the validity of four applied distribution functions for precipitation (P), D (D = P − PET), WER, Z, and $Z_{snow}$ across different climate zones. These climate zones were classified on the basis of the Aridity Index (AI), which is computed by the ratio of the long-term mean annual PET to P (AI = PET/P). A drier local climate condition corresponds to a higher AI value. Therefore, regions with AI

equal to 0.76, 1.45, 2.5, 4.4, and 79.9 represent humid, sub-humid, semi-arid, arid, and extreme arid regions.

As shown in Figure R5, we tested four possible three-parameter distributions to model the P, D, WER, Z, and $Z_{snow}$ values at different climate zones, including Pearson III, log-logistic, lognormal, and general extreme value (GEV) distribution. For this purpose, the L-moment ratio diagram was used to examine the performance of different distributions because it allows comparing the empirical frequency distribution of P, D, WER, Z, and $Z_{snow}$ time series at different climate regions with several theoretical distributions. The results show the log-logistic distribution generally fits the P, D, WER, Z, and $Z_{snow}$ data the best across multiple climate zones. Therefore, the log-logistic distribution can reasonably be adopted for standardizing the P, D, WER, Z, and $Z_{snow}$ data in calculating the SPI, SPEI, SWI, SZI, and $SZI_{snow}$.

[Figure]

[Figure]

Figure R5. Empirical and modeled values (f(x) and F(x) or probability density function and cumulative distribution function, respectively) using the Pearson III, log-logistic, lognormal, and general extreme values (Gen. Ext. Value) distributions of the P, D, WER, Z, and $Z_{snow}$ series at monthly scale for deriving the SPI, SPEI, SWI, SZI, and $SZI_{snow}$.

**Minor comments**

**1) Lines 19-20, 22**

**Some numbers keep two decimals, and some do not.**

Reply: The number in line 22 has been revised and consistent with other numbers. In the revised manuscript (Page 1, Line 22), this line was changed to "*of which 68.38% had a duration longer than 6 months.*"

**2) Lines 99-101**

**Show some reasons to explain why the better performance of GLDAS-2 compared to that of GLDAS-1?**

Reply: Various studies have shown that the GLDAS-2 has a better performance than the GLDAS-1. These studies pointed out several causes for such better performance of GLDAS-2, and here we summarized the causes as follows. GLDAS-1 has serious discontinuity issues in its forcing data. Especially, GLDAS-1 precipitation data have larger errors in 1996, and the snowfall amount has approximately doubled after 2000. These discontinuity issues are primarily attributed to several switches of the forcing data of GLDAS-1. In contrast, GLDAS-2 precipitation has much better temporal continuity than GLDAS-1 precipitation, as GLDAS-2 uses the bias-corrected Princeton meteorological forcing dataset. Additionally, evaluations over the high-altitude regions indicate that GLDAS-2 models perform better than the GLDAS-1 models in simulating runoff. This is mainly because the GLDAS-2 models integrate glacier melt runoff in the simulation, whereas the GLDAS-1 models demonstrate no glacier melt runoff.

Following the comment, we have added these causes to explain the better performance of GLDAS-2 compared to GLDAS-1 in the revised manuscript (Page 5, Lines 105-109) as follows:

*This is mainly attributed to that GLDAS-1 has serious discontinuity problems in its meteorological forcing dataset due to switches in its forcing data. In contrast, GLDAS-2 has a better temporal continuity, using the bias-corrected Princeton*

*meteorological forcing dataset. Additionally, evaluations for the high-altitude regions indicate that GLDAS-2 performs better in streamflow simulation because GLDAS-2 considers streamflow from glacier melt in its simulation, but the GLDAS-1 did not.*

**3) Line 140**

**I did not quite understand why the WER is regarded as a comprehensive drought indicator? What is a comprehensive drought indicator? Is there any definition for it? Please give more information.**

Reply: The residual water–energy ratio (WER) was first suggested by Liu et al. (2017). As mentioned in the manuscript, the ratio of the residual available water to the residual energy (PET−ET) is relatively low (large) during drought (wet) events relative to normal conditions. Defining this ratio as WER = (P − ET)/(PET − ET), Liu et al. (2017) proposed a method for examining the response of the surface water-energy fluxes to drought based on WER.

Droughts are generally classified into three categories (meteorological, hydrological, and agricultural droughts) based on their physical characteristics. Nevertheless, these different categories make it difficult to objectively quantify drought features. Moreover, different types of droughts are considered to be interchangeable. Drought can convert from one type to another as it evolves in time and space. Zhang et al. (2019) thus recommend employing the WER, a comprehensive drought indicator, to comprehensively describe drought through a water-energy balance perspective. We can objectively and easily quantify drought across different spatial scales through such a perspective.

**4) Line 155**

**Did you evaluate the capacity of your dataset across different climate zones? I did not see these mentioned climate zones in the following sections of the manuscript. I guess you wanted to say that the evaluation was conducted over different geographical parts of the world. Please clarify this.**

Reply: This is a typo. This line has been corrected in the revised manuscript (Page 12, Line 238) as "*multiscalar drought across different geographical parts of the world*."

**5) Lines 221-222**

**What is the name of dimensions in your 3D and 2D dataset? Please clarify this.**

Reply: The names of dimensions in the 3D dataset are month, latitude, and longitude. The names of dimensions in the 2D dataset are latitude and longitude. This sentence has been modified in the revised manuscript (Page 12, Lines 247-248) as follows:

*The SAD method firstly uses a monthly three-dimensional (3D, $month \times latitude \times longitude$) gridded drought index dataset to identify two-dimensional (2D, $latitude \times longitude$) drought clusters in each time step.*

**6) Line 273**

**It seems this content has been repeated in the main text. Delete it to make the caption more concise.**

Reply: This content has been removed from the revised manuscript (Page 15, Line 309) following the comment.

**7) Line 353**

**Is the geographical extent of Oceania equal to that of Australia? Give an exact definition.**

Reply: The geographical extent of Oceania is not equal to Australia. This study defines Oceania as Australia, New Zealand, Papua New Guinea, and the Pacific Islands. This content has been added in the revised manuscript (Page 21, Lines 391-392) as follows:

*Here Oceania is defined as Australia, New Zealand, Papua New Guinea, and the Pacific Islands.*

**8) Why wasn't Greenland included in Figure 6? It seems all the Greenland are missing values.**

Reply: Figure 6 in the original manuscript has been updated as Figure 7 in the revised manuscript. Greenland was excluded from our study because about 80% of its surface is ice-capped. In addition, Greenland has a small population of nearly 56,100 (in 2016) on an area of 2,166,086 km², making Greenland the least densely populated place on earth. Thus, the influence of droughts on this island is minimal, and the droughts in Greenland were not of interest for this study.

Following the comment, the reason for the exclusion of Greenland has been added in the revised manuscript (Page 19, Lines 361-362) as follows:

***Our study excluded Greenland due to its sizeable ice-capped area about 80% of the island.***

**9) When you did the SAD analysis, how to process the drought over the Sahara Desert?**

Reply: When we did the SAD analysis, drought events were tracked through time by searching for overlapping grid cells between clusters at contiguous time steps. Clusters were allowed to propagate into the Sahara until their centroids fell within the specified domain that is outside the Sahara. Droughts whose centroids fell within the Sahara Desert (20°N–25°N, 17°W–34°E) were removed from our analysis since droughts in these regions were not of interest for this study.

Following the comment, how to process droughts over the Sahara in the SAD analysis has been added in the revised manuscript (Pages 12-13, Lines 253-256) as follows:

***Additionally, droughts in the Sahara Desert were not concerned in our study. During the SAD analysis, clusters were allowed to propagate into the Sahara (20°N–25°N, 17°W–34°E), and these clusters would be retained if their centroids fell outside the Sahara Desert. In contrast, drought clusters were discarded if their centroids locate in the Sahara Desert.***

**Technical comments:**

**1) Figure 1: The font size of the description to compare the strength and weakness of each index is small.**

Reply: In the revised manuscript, we have enlarged the font size of the description to compare the strength and weakness in Figure 1 (Page 4).

**2) Figure 4: The font should be the same, use one type.**

Reply: We have revised the font and use the same font in this figure in the revised manuscript (Page 17).

**3) Figure 6: The stippling seems unclear.**

Reply: We have increased the resolution of this figure in the revised manuscript (Page 19).

**4) Figure 9: Please supply information on the geographic coordinate system used in this figure. It can help others to compare their results with yours.**

Reply: This figure uses the geographic coordinate system (latitude and longitude). We have added this information into its caption, and this figure has been updated as Figure 10 in the revised manuscript (Page 26, Lines 455-456).

**5) Figure S2: Adjust the minimum value of the legend. There is no grid with a SZIsnow value less than -4.0.**

Reply: The minimum level of the SZIsnow was set as -4.0 in Figure S2, since we wanted to keep the legend of Figure S2 to be consistent with that in Figure 10 in the revised manuscript.

**Comments for the dataset**

**1) Table 1 in the metadata file should be kept the same as Table 2 in the manuscript.**

Reply: We will update Table 1 in the metadata file at the National Tibetan Plateau/Third Pole Environment Data Center (http://data.tpdc.ac.cn/en/disallow/b039fde6-face-4d24-af45-d238a6af18b7/).

**2) Add information relative to the size of decompressed files in the metadata file.**

Reply: We will add the below information relative to the size of decompressed files in the metadata file at the National Tibetan Plateau/Third Pole Environment Data Center (http://data.tpdc.ac.cn/en/disallow/b039fde6-face-4d24-af45-d238a6af18b7/):

*The size of each zipped file is 14.5 GB (gigabyte), and each file in a zipped file is 2.43 GB. The total size of all the decompressed files is 116 GB.*

**3) I recommend providing a thumbnail of your dataset in the metadata file.**

Reply: We have added a thumbnail of our dataset in the metadata file at the National Tibetan Plateau/Third Pole Environment Data Center (http://data.tpdc.ac.cn/en/disallow/b039fde6-face-4d24-af45-d238a6af18b7/) as follows:

[Figure]

Figure R6. Spatial distribution of the $SZI_{snow}$ on July 1992 at a 6-month timescale.

**4) If possible, provide some scripts for potential readers to plot your data.**

Reply: We will provide two scripts for potential readers to plot the $SZI_{snow}$ data in the metadata file. The first script is coded based on Python programming language, and another is coded based on the NCAR Command Language (NCL).

**Referee #4:** https://doi.org/10.5194/essd-2021-399-RC4

**Overall comments**

**This manuscript developed a new global monthly drought index dataset with multi-types and multi-scales, SZIsnow. The drought index SZIsnow incorporates different physical water–energy processes with snow process. The dataset also was comprehensively evaluated by different drought types, different spatial scales. The drought index SZIsnow and SZI are compared in different regions.**
**The dataset can serve as a valuable resource for drought studies. The paper is scientifically sounding. The topic well fits the scope of this special issue. The manuscript is well written and logically organized and the dataset was easy to access. However, some concerns still need to be addressed and make it clearer for the readers before publication. Below are my several comments.**

Reply: We appreciate the referee's comments on our work.

**General Comments**

**1. The title used "standardized moisture anomaly index", however, "drought index" is used more in the text. Need to consider a better title to attract the interest of the potential data users.**

Reply: We thank the referee for this excellent comment. The title has been changed to "*A global drought dataset of standardized moisture anomaly index incorporating snow dynamics (SZI$_{snow}$) from 1948 to 2010*" in the revised manuscript.

**2. The abstract does not show the spatial resolution of your dataset. It is an essential parameter for reader and data user.**

Reply: We cannot agree more with this comment. The spatial resolution of a dataset is an essential parameter for readers and data users. The spatial resolution of our proposed dataset is 0.25 degrees. This content has been added in the revised manuscript (Page 1, Line 12) as follows:

*Here, we present a global monthly drought dataset with a spatial resolution of 0.25°*
*from 1948 to 2010 based on a multitype and multiscalar drought index, the*
*standardized moisture anomaly index incorporating snow dynamics (SZIsnow), driven*
*by systematic fields from an advanced data assimilation system.*

**3. In the Line 20, "Our results also show that the SZIsnow dataset successfully
captured the largescale drought events that occurred across the world; there were
525 drought events with an area larger than 500,000 km$^2$ globally during the study
period, of which nearly 70% had a duration longer than 6 months." What is the
accuracy rate of this product? How to evaluate this more reasonable? The product
capture all the drought events? Is its capture rate 100%?**

Reply: We thank the referee for this valuable comment. We recognize that the proposed
product cannot capture all the largescale drought events, and the capture rate is not
100%. The accuracy of drought assessment is a challenge for drought study, primarily
because there did not exit an indicator to quantify drought directly. The lack of long-
term observation is also the main reason for the difficulty of drought assessment. In
addition, incomplete model structure, forcing data biases, and biases in parameter
estimation in the forcing dataset (GLDAS-2) of the SZI$_{snow}$ can lead to the inaccuracy
of our results. Furthermore, the clustering algorithm of the SAD method allows for the
merging of two or more sub-droughts into a drought. Although the smooth and
continuous movement of the drought clusters hints at some common underlying
mechanisms, it is sometimes difficult to interpret sub-droughts as a single event due to
likely different forcing mechanisms behind them. From the above considerations, it is
hard to give an accurate rate of the captured droughts.

To the best of the authors' knowledge, the reasonable approach for evaluating our
results is the comparison with the published paper in the science community. As shown
in Section 4.3.2, we compared our captured events with those documented by other
studies for each continent. Although these documented drought events were identified
based on different drought indices, these comparisons indicate that our captured
drought events are broadly aligned with findings from previous research. For example,

Sheffield et al. (2009) pointed out that there have been 296 droughts greater than 500,000 km$^2$ globally from 1950 to 2000, with a dataset of soil moisture from simulations using the variable infiltration capacity model. This number is 311 in our result during the same period.

We recognize that the sentence in Line 20 is not an appropriate description of our results. Therefore, following the comment, this sentence has been corrected in the revised manuscript (Page 1, Lines 20-23) as follow:

*Our results also indicate that the SZI$_{snow}$ dataset can be employed to capture the largescale drought events that occurred across the world. Our analysis shows there were 525 drought events with an area larger than 500,000 km$^2$ globally during the study period, of which 68.38% had a duration longer than 6 months.*

**4. In the Figure 1. I did not see the description of scPDSI in the manuscript.**

Reply: As it is known, the self-calibrated PDSI (scPDSI) is developed based on the original PDSI and accounts for all the constants contained in PDSI. The scPDSI includes a methodology in which the constants are calculated dynamically based upon the characteristics present at each station location. Thus, the scPDSI can generate more representative model constants at temporal and spatial scales. In Figure 1, as the methodology is not significantly different from PDSI, it has the same issues as the PDSI in terms of the fixed temporal scale and poor performance over snow-covered areas. Therefore, we did not mention scPDSI in the manuscript.

Following the comment, we have added the below content in the revised manuscript (Page 2, Lines 49-52):

*In addition, the self-calibrated PDSI (scPDSI) can compute dynamically the constants in PSDI on the basis of the characteristics at each interested location, producing more representative model constants. However, the scPDSI has the same issues as the PDSI in terms of the temporal scale and performance over snow-covered areas.*

**5. I suggest adding the description of the advantages of the SZIsnow in the figure 1.**

Reply: Following the comment, we have added the description of the advantages of the SZI$_{snow}$ in Figure 1 of the revised manuscript (Page 4):

[Figure]

*Figure 1. Development path of the SZI$_{snow}$. Dark green boxes denote the strengths of each drought index, while pink boxes denote the weaknesses of each drought index. The top row shows indices that can only account for one type of drought, with three indices listed for each type of drought. The second row shows indices that can account for multiple types of drought. Full names of the listed indices are shown in Table S1.*

**6. The GLDAS-2 data provide the variables to calculate the SZIsnow from 1948-2010. GLDAS-2.1 is one of two components of the GLDAS Version 2 (GLDAS-2) dataset, from 2000 to present. Is it possible to use GLDAS-2.1 to extend the time coverage of the product?**

Reply: We thank the referee for this valuable suggestion. The time coverage of a drought index product is essential for drought study. The GLDAS-2.0 and GLDAS-2.1 are two components of the GLDAS Version 2 dataset (GLDAS-2). Although the GLDAS-2.0 is analogous to GLDAS-2.1, there are some differences in their meteorological forcing datasets. Thus, the performance of these two components might be different for a specific region. For example, a previous study found that different performances of GLDAS2.0 and GLDAS2.1 exists in runoff simulations over the Tibetan Plateau, which may be due to their different uncertainty in the forcing data (Qi et al., 2018). Moreover, the moisture anomaly ($Z_{snow}$) in $SZI_{snow}$ can be aggregated at different temporal scales, causing the former months have influences on the latter months. An abrupt jump can be found in the time series of $SZI_{snow}$ when we connect these two different forcing datasets, which will inevitably introduce systematic bias in the $SZI_{snow}$. Based on the above considerations, we temporarily did not extend the time coverage of the $SZI_{snow}$ dataset using the GLDAS-2.1 product. In further work, this suggestion will be taken full account. We will appraise these two components and develop an approach to balance their differences, extending the time coverage of the $SZI_{snow}$ dataset.

Qi, W., Liu, J., & Chen, D. (2018), Evaluations and Improvements of GLDAS2.0 and GLDAS2.1 Forcing Data's Applicability for Basin Scale Hydrological Simulations in the Tibetan Plateau, *Journal of Geophysical Research: Atmospheres*, 123(23), 13,128-113,148, https://doi.org/10.1029/2018JD029116.

**7. Section 2.4 "Metrics for the SZIsnow evaluation" is not the data. It is an accuracy assessment method, not the data description. Is it more appropriate to move this part to the section 3.**

Reply: Following the comment, we have moved this section to Section 3 as Section 3.2 in the revised manuscript (Page 12, Lines 232-238).

**8. Line 164 "The prominent improvement of the SZIsnow is that it accounts for the influence of snowfall on hydrological processes, which was completely ignored in the SZI (Zhang et al., 2019; Zhang et al., 2015)." Line 171 "Both the soil moisture storage and snow storage are considered as reservoirs in the SZIsnow, which is different from the SZI that solely considered the former." This section is to discuss how to produce the SZIsnow. The difference between SZI and SZIsnow should be placed in the validation section.**

Reply: We should focus on how to produce the $SZI_{snow}$, instead of comparing the SZI and $SZI_{snow}$, since Section 3.1 discusses the derivation of the $SZI_{snow}$. We checked this section thoroughly to avoid a similar problem following the comment. Since similar content about the difference between SZI and $SZI_{snow}$ already exits in the validation section, all the kind of sentences have been removed from Section 3.1.1.

**9. I suggest a procedure flowchart describing the production and validation of SZIsnow. It would be better that the advantages of the SZIsnow are mentioned in the figure. The flowchart can facilitate users to understand the dataset.**

Reply: We cannot agree more with this comment. A procedure flowchart can facilitate users to understand the production and validation of the $SZI_{snow}$ dataset. Following the comment, we have added the below figure (Page 9) and its corresponding text (Page 8, Lines 178-185) in Section 3 of the revised manuscript:

*We provide a procedure flowchart as shown in Figure 3 to show the production and validation of the $SZI_{snow}$. There are four steps for $SZI_{snow}$ production: hydrologic accounting, climatic coefficients, water demand, and standardization. Hydrologic accounting is to calculate the monthly six components relevant to the local water budget; Climatic coefficients are the weighting factors of these components for the calculation of the local water demand; The local water demand in the $SZI_{snow}$ is represented by the precipitation that is climatically appropriate for existing*

*conditions (CAFEC, referred as $\widehat{P}_{snow}$); The last step is the standardization of the moisture anomaly ($Z_{snow}$), which is the difference between the actual precipitation (rainfall and snowfall). After achieving the global SZI$_{snow}$ dataset, its ability to identify different types of drought can be validated not only at basin scale, but also across different regions worldwide, especially snow-covered regions, at grid scale.*

[Figure]

*Figure 3. The procedure flowchart describing the production and validation of SZI$_{snow}$. Variables derive the SZI$_{snow}$ from the GLDAS-2 (or other LSM and DAS). The production of SZI$_{snow}$ includes four steps. The SZI$_{snow}$ is validated at basin scale for three types of drought and at grid scale across different regions worldwide, respectively. The cloud-shape annotation shows the advantages of the SZI$_{snow}$.*

---

## Author Response (AR2)

"A global drought dataset of standardized moisture anomaly index incorporating snow dynamics (SZI$_{snow}$) from 1948 to 2010" by Tian et al.

Response to Editor

We thank the Editor for your time and effort in handling and reviewing our manuscript. Here we address each of your comments and clarify the corresponding changes in the revised manuscript. The original comments are in **black bolded font**, our responses are in blue, and manuscript changes are in *bold italic*.

**Comments to the author:**

**This paper proposed a new global drought product considering a relatively comprehensive extent of the hydrometeorological variables associated with snow information. This is a great contribution to the drought analysis at a global scale. I have several comments as follows:**

**1. Line 519: at the following URL: http://data.tpdc.ac.cn/en/disallow/b039fde6-face-4d24-af45-d238a6af18b7/. This data link seems to be only a temporary link. Can it be changed into a formal link?**

Reply: We have removed the temporary link and added a formal link in the revised manuscript as follows (Page 28, Lines 519-520):

*The SZI$_{snow}$ dataset can be downloaded from this data center at the following URL: http://data.tpdc.ac.cn/en/data/b039fde6-face-4d24-af45-d238a6af18b7/.*

**2. Since you have put the data in National Tibetan Plateau/Third Pole Environment Data Center, you are welcome to cite the relevant introduction papers into the articles as: https://doi.org/10.1175/BAMS-D-21-0004.1 and https://doi.org/10.1175/BAMS-D-19-0280.1**

Reply: We have added the two relevant introduction papers into the revised manuscript as follows (Page 28, Lines 516-519):

*In addition, we also published the dataset to the National Tibetan Plateau/Third Pole Environment Data Center, which has been accredited by the Earth System Science Data, and specializes in collecting, integrating, and publishing geoscientific data on and surrounding the Tibetan Plateau and the three poles (Li et al., 2020; Pan et al., 2021).*